# Rats exhibit similar biases in foraging and intertemporal choice tasks

Gary A Kane[1,2]*, Aaron M Bornstein[1,3], Amitai Shenhav[4], Robert C Wilson[5], Nathaniel D Daw[1], Jonathan D Cohen[1]

[1]Department of Psychology, Princeton Neuroscience Institute, Princeton University, Princeton, United States; [2]Rowland Institute at Harvard, Harvard University, Cambridge, United States; [3]Department of Cognitive Sciences, Center for the Neurobiology of Learning and Memory, University of California, Irvine, Irvine, United States; [4]Department of Cognitive, Linguistic and Psychological Sciences, Carney Institute for Brain Science, Brown University, Providence, United States; [5]Department of Psychology, Cognitive Science Program, University of Arizona, Tucson, United States

**Abstract** Animals, including humans, consistently exhibit myopia in two different contexts: foraging, in which they harvest locally beyond what is predicted by optimal foraging theory, and intertemporal choice, in which they exhibit a preference for immediate vs. delayed rewards beyond what is predicted by rational (exponential) discounting. Despite the similarity in behavior between these two contexts, previous efforts to reconcile these observations in terms of a consistent pattern of time preferences have failed. Here, via extensive behavioral testing and quantitative modeling, we show that rats exhibit similar time preferences in both contexts: they prefer immediate vs. delayed rewards and they are sensitive to opportunity costs of delays to future decisions. Further, a quasi-hyperbolic discounting model, a form of hyperbolic discounting with separate components for short- and long-term rewards, explains individual rats' time preferences across both contexts, providing evidence for a common mechanism for myopic behavior in foraging and intertemporal choice.

DOI: https://doi.org/10.7554/eLife.48429.001

*For correspondence:
gkane@rowland.harvard.edu

**Competing interests:** The authors declare that no competing interests exist.

## Introduction

Serial stay-or-search problems are ubiquitous across many domains, including employment, internet search, mate search, and animal foraging. For instance, in patch foraging problems, animals must choose between an immediately available opportunity for reward or the pursuit of potentially better but more distal opportunities. It is typically assumed that animals seek to maximize the long-term average reward (net of cost) rate, as a proxy for reproductive fitness. The optimal behavior for maximizing this currency in foraging tasks, described by the Marginal Value Theorem (MVT; *Charnov, 1976*), is to choose the immediately available opportunity if it provides a reward rate greater than the average reward rate across all alternative options, which includes the costs of accessing those options. Animals tend to follow the basic predictions of long-term reward maximization: they are generally more likely to pursue opportunities for larger vs. smaller rewards and, if the cost of searching for alternatives is greater, they are more likely to pursue opportunities for smaller rewards (*Stephens and Krebs, 1986*; *Constantino and Daw, 2015*; *Hayden et al., 2011*; *Kane et al., 2017*).

Although animal behavior follows the basic predictions of optimal foraging behavior described by MVT, in the majority of studies across a variety of species, including humans, non-human primates, and rodents, animals exhibit a consistent bias towards pursuing immediately available rewards

**eLife digest** Often decisions have to be made on whether to stick with a resource or leave it behind to search for a better alternative. Should you book that hotel room or continue looking at others? Is it time to start searching for a new job, or even for a new partner? Animals face similar 'stick or twist' decisions when foraging for food. Knowing how to maximize the amount of food you obtain is key to survival. Studies have shown that most animals tend to stick with a food source for a little too long, a phenomenon known as 'overharvesting'.

To find out why, Kane et al. designed carefully controlled experiments to compare foraging behavior in rats to another form of decision-making, known as intertemporal choice. The latter involves choosing between a small reward now versus a larger reward later. Given this choice, most rats opt to receive a smaller reward now rather than wait for the larger reward. This suggests that rats value rewards available in the future less than rewards they can get immediately.

Kane et al. showed that this preference for short-term rewards can also explain why rats overharvest in foraging scenarios. By leaving one food source to go in search of another, rats must put up with a delay before they can access the new food supply. This delay, due to the time required to travel and search, reduces the value of the future reward. As a result, rats are more likely to stick with their current food source, even though leaving it would yield a greater reward in the long run.

These findings in rats raise important questions about the mechanisms that lead to biases in thinking, and how factors like changes in the environment or specific disease states can influence these biases.

DOI: https://doi.org/10.7554/eLife.48429.002

relative to predictions of MVT, often referred to as 'overharvesting' (*Constantino and Daw, 2015*; *Hayden et al., 2011*; *Kane et al., 2017*; *Nonacs, 2001*; *Kolling et al., 2012*; *Shenhav et al., 2014*; *Wikenheiser et al., 2013*; *Carter and Redish, 2016*). Prior studies have proposed two explanations for overharvesting: subjective costs, such as an aversion to rejecting an immediately available reward (*Wikenheiser et al., 2013*; *Carter and Redish, 2016*); and nonlinear reward utility or diminishing returns, by which larger rewards are not perceived as proportionally larger than smaller rewards (*Constantino and Daw, 2015*). But these hypotheses have never been systematically compared in a set of experiments designed to directly test their predictions. Furthermore, according to these rate-maximizing hypotheses, the perceived value of rewards does not differ between situations in which the delays occur before or after reward is received. In this respect, the predictions made by these hypotheses (which are still grounded in a core assumption that animals attempt to maximize the long-term reward rate) are not compatible with an otherwise seemingly similar bias that is widely observed in standard intertemporal choice tasks (also referred to as delay discounting or self-control tasks): a preference for smaller, more immediate rewards over larger, delayed rewards (*Ainslie, 1992*; *Kirby, 1997*).

The preference for more immediate rewards in intertemporal choice tasks is commonly explained in one of two ways, both assuming that animals choose as though they were optimizing a different currency than long-term reward rate: temporal discounting or short-term rate maximization. According to temporal discounting, the perceived value of a future reward is discounted by the time until its receipt. Temporal discounting can arise even when maximizing the long-term reward rate, for certain environments. In particular, discounting can be adaptive in unstable environments — if the environment is likely to change before future rewards can be acquired, it is appropriate to place greater value on more predictable rewards available in the near future. Under this hypothesis, and the further assumption that expected rewards disappear at a constant rate, a long-term reward rate maximizer would discount rewards exponentially in their delay (*Gallistel and Gibbon, 2000*; *Kacelnik and Todd, 1992*). However, animal preferences typically follow a hyperbolic-like form: the rate of discounting is steeper initially and decreases over time (*Gallistel and Gibbon, 2000*; *Kacelnik and Todd, 1992*; *Thaler, 1981*). This yields inconsistent time preferences or preference reversals: an animal may prefer to wait longer for a larger reward if both options are distant, but will change their mind and prefer the smaller reward as the time to both options draws near

(*Ainslie, 1992*; *Kirby, 1997*; *Gallistel and Gibbon, 2000*; *Kacelnik and Todd, 1992*). Recent theoretical work has shown that hyperbolic time preferences may arise from imperfect foresight — if the variance in predicting the timing of future outcomes increases with the delay to the outcome, a long-term reward rate maximizer would exhibit hyperbolic time preferences (*Gabaix and Laibson, 2017*). Similarly, short-term maximization rules predict that animals seek to maximize reward over shorter time horizons; this may also be motivated as an approximation to long-term reward maximization as it may be difficult to accurately predict all future rewards (*Stephens, 2002*; *Stephens et al., 2004*). Along similar lines, *Namboodiri et al. (2014)* argues that, rather than maximizing long-term reward rate into the future, animals may select options that maximize reward rate up to the current point in time or due to environmental factors (e.g. non-stationarity) or biological constraints (e.g. computational constraints), over a finite interval of time. Just as hyperbolic discounting may arise from imperfect foresight (*Gabaix and Laibson, 2017*), maximizing reward rate over shorter time horizons predicts hyperbolic time preferences (*Namboodiri et al., 2014*).

An alternative explanation for the preference for immediate rewards in intertemporal choice tasks is that animals simply underestimate the duration of post-reward delays; that is, delays between receiving reward and making the next decision (*Pearson et al., 2010*; *Blanchard et al., 2013*). Typically, in intertemporal choice tasks, a variable post-reward delay is added to ensure that the overall amount of time for each trial is equal, regardless of the reward size or the duration of the pre-reward delay. It has been argued that it may be difficult for animals to learn these variable delays, and thus, animals may fail to consider the full duration of the delay in their decision process. Consequently, animals will perceive that it takes less time to acquire the smaller, more immediate reward and overestimate the reward rate for choosing this option. Consistent with this hypothesis, providing an explicit cue for the duration of the post-reward delay or increasing its salience by providing a small reward at the end of the post-reward delay reduces temporal discounting (*Pearson et al., 2010*; *Blanchard et al., 2013*).

Despite the similarities between overharvesting and the preference for more immediate rewards in intertemporal choice tasks, prior attempts to use temporal discounting and/or short-term rate maximization functions fit to intertemporal choice data to predict foraging behavior have failed (*Carter et al., 2015*; *Carter and Redish, 2016*; *Blanchard and Hayden, 2015*). In these studies, animals are typically closer to long-term rate maximization in foraging tasks than in intertemporal choice tasks. This has been taken as further evidence that, while animals have a good understanding of the structure of foraging tasks, they struggle to understand the structure of intertemporal choice tasks (i.e. they fail to incorporate post-reward delays into their decision process; *Blanchard and Hayden, 2015*). However, there are two additional possibilities for why intertemporal choice models have failed to predict foraging behavior. First, models of intertemporal choice tasks usually consider rewards for the current trial and not rewards on future trials since, in these tasks, reward opportunities on future trials are often independent of the current decision. This is not true of foraging tasks, in which future opportunities for rewards depend on the current decision. Thus, this difference in decision horizon may make it difficult to explain foraging data using discounting models fit to intertemporal choice data. Second, these studies have only examined standard, single-parameter exponential and hyperbolic discounting functions. More flexible forms of temporal discounting that produce different patterns of hyperbolic time preferences have never been tested in these contexts. More flexible discounting functions include constant sensitivity discounting, by which rewards in the distant future are discounted less than rewards in the near future due to a bias in time estimation — agents become less sensitive to longer time delays (*Ebert and Prelec, 2007*; *Zauberman et al., 2009*); additive-utility discounting, by which the utility of a reward, not its value, is discounted (*Killeen, 2009*); or quasi-hyperbolic discounting, which has separate terms, or different discount rates, for short- or long-term rewards rewards that correlate with activity in limbic and fronto-parietal networks respectively (*Laibson, 1997*; *McClure et al., 2004*; *McClure et al., 2007*).

In the present study, we found that rats exhibit similar time preferences in foraging and intertemporal choice tasks and that time preferences in both tasks can be explained by a quasi-hyperbolic discounting model that, in both contexts, considers future rewards. Rats were tested in a series of patch foraging tasks and an intertemporal choice task. In foraging tasks, they followed the basic predictions of long-term rate maximization: they stayed longer in patches that yielded greater rewards and when the cost of searching was greater. But under certain conditions, they violated these predictions in a manner consistent with time preferences: they stayed longer in patches when given

larger rewards with proportionally longer delays, and they exhibited greater sensitivity to pre- vs. post-reward delays. Similarly, in an intertemporal choice task, rats exhibited greater sensitivity to pre- vs. post-reward delays. Using these data, we tested several models to determine if temporal discounting or biases in time perception, such as insensitivity to post-reward delays, could explain rats' behavior across tasks. One model, a quasi-hyperbolic discounting model (*Laibson, 1997*; *McClure et al., 2007*), provided the best fit to rat behavior across all experiments. Furthermore, the quasi-hyperbolic discounting model proved to be externally valid: discounting functions fit to foraging data provided as good a fit to intertemporal choice data as discounting functions fit directly to intertemporal choice data for some rats. These findings suggest that rats exhibit similar biases in the two tasks, and quasi-hyperbolic discounting may be a common mechanism for suboptimal decision-making across tasks.

## Results

### Rats consider long-term rewards, but exhibit a bias in processing pre- vs. post-reward delays

Long Evans rats (n = 8) were tested in a series of patch foraging tasks in operant conditioning chambers (*Kane et al., 2017*). To harvest reward (10% sucrose water) from a patch, rats pressed a lever on one side of the front of the chamber (left or right) and reward was delivered in an adjacent port. After a post-reward delay (inter-trial interval or ITI), rats again chose to harvest a smaller reward or to leave the patch by nose poking in the back of the chamber. A nose poke to leave the patch caused the harvest lever to retract and initiated a delay to control the time to travel to the next patch. After the delay, the opposite lever extended (e.g. if the left lever was extended previously, the right lever would be extended now), and rats could then harvest from (or leave) this replenished patch (*Figure 1—figure supplement 1*).

In four separate experiments, we manipulated different variables of the foraging environment: (i) in the 'Travel Time Experiment,' a 10 s vs. 30 s delay was imposed between patches, (ii) in the 'Depletion Rate Experiment,' reward depleted at a rate of 8 vs. 16 μL per harvest, (iii) in the 'Scale Experiment,' the overall magnitude of rewards and delays was varied, such that in one condition, the size of rewards and length of delays was twice that of the other. (iv) Finally, in the 'Pre-vs-Post Experiment,' the placement of delays was varied, such that the total time to harvest reward remained constant, but in one condition there was no pre-reward delay and ~13 s post reward delay, and in the other there was a 3 s pre-reward delay and ~10 s post-reward delay. Parameters for each experiment

**Table 1.** Parameters for each of the first four foraging experiments.
Harvest time = time to make a decision to harvest + pre-reward delay + post-reward delay (inter-trial interval). To control reward rate in the patch, the post-reward delay was adjusted relative to the decision time to hold the harvest time constant.

| Experiment | Condition | Start Reward | Depletion Rate | Pre-Reward Delay | Harvest Time | Travel Time |
|---|---|---|---|---|---|---|
| travel time | 10 s | 60, 90, or 120 μL[†] | −8 μL | 0 s | 10 s | 10 s |
| | 30 s | | | | | 30 s |
| depletion rate | −8 μL | 90 μL | −8 μL | 0 s | 12 s | 12 s |
| | −16 μL | | −16 μL | | | |
| scale | 90 μL/10 s | 90 μL | −8 μL | 0 s | 10 s | 10 s |
| | 180 μL/20 s | 180 μL | −16 μL | | 20 s | 20 s |
| handling time | 0 s | 90 μL | −8 μL | 0 s | 15 s | 15 s |
| | 3 s | | | 3 s | | |
| post-reward delay* | 3 s | 90 μL | −8 μL | 0 s | 5–8 s** | 10 s |
| | 12 s | | | | 13–16 s** | |

[†]Rats encountered all three patch types in both conditions.

*One group of rats (n = 8) was tested on the first four experiments, but a separate group (n = 8) was tested on this final foraging experiment.

**In this experiment, the harvest time was not held constant — the post-reward delay was always 3 s or 12 s regardless of the time to make a decision.

DOI: https://doi.org/10.7554/eLife.48429.006

are shown in *Table 1*. For each condition within each experiment, rats were trained for 5 days and tested for an additional 5 days; all behavioral data presented is from the 5 test days. The order of conditions within each experiment was counterbalanced across rats. Every patch visit was included for analysis; mixed effects models were used to examine the effect of task condition on the number of trials spent in each patch. Random intercepts and random slopes for the effect of task condition were used to group observations within each rat. To compare rat behavior to the optimal behavior in each condition, a mixed effects model was used to test the effect of task condition on the difference between the number of trials spent in each patch and the optimal number of trials for that patch, with random intercepts and slopes for each rat. For this mixed effects model, an intercept of zero indicates optimal performance, and the slope indicates the change in behavior relative to the optimal behavior between conditions (see Materials and ethods for additional detail).

The Travel Time Experiment was designed to test the two main predictions of MVT: (i) that animals should stay longer in patches that yield greater rewards and (ii) animals should stay longer in all patches when the cost of traveling to a new patch is greater. In this experiment, rats encountered three different patch types within sessions, which started with varying amount of reward (60, 90, or 120 µL) and depleted at the same rate (8 µL/harvest). The delay between patches was either 10 s or 30 s; each travel time delay was tested in its own block of sessions and the order was counterbalanced across rats, with a range of 87–236 patches visited per condition per rat. As predicted by MVT, rats stayed for more trials in patch types that started with larger reward volume ($\beta$ = 118.091 trials/mL, SE = 1.862, t(2490.265) = 63.423, p < .001), indicating that rats considered reward across future patches. Rats also stayed longer in all patch types when time between patches was longer ($\beta$ = 1.893 trials, SE = 0.313, t(118.839) = 6.040, p < .001; *Figure 1A*), indicating sensitivity to opportunity costs. However, rats uniformly overharvested relative to predictions of MVT ($\beta_{rat-MVT}$ = 3.396 trials, SE = 0.176, t(6.960) = 19.269, p < .001). The degree to which rats overharvested was not significantly different between the 10 s and 30 s travel conditions ($\beta_{10\ s-30\ s}$ = 0.304 trials, SE = 0.155, t (7.3857) = 1.964, p = 0.088).

The Depletion Rate Experiment tested another critical variable in foraging environments: the rate of reward depletion within a patch. Quicker reward depletion causes the local reward rate to deplete to the long-run average reward rate quicker, thus MVT predicts earlier patch leaving. Within sessions, rats encountered a single patch type (starting volume of 90 µL) that depleted at a rate of either 8 or 16 µL/trial, tested in separate sessions and counterbalanced, with a range of 152–283 patches visited per condition per rat. As predicted by MVT, rats left patches earlier when patches depleted more quickly ($\beta$ = 2.589 trials, SE = 0.155, t(7.000) = 16.75, p < .001; *Figure 1B*). But, again, rats stayed in patches longer than is predicted by MVT ($\beta_{rat-MVT}$ = 2.005 trials, SE = 0.134, t (7.004) = 14.97, p < .001). Rats overharvested to a greater degree in the 8 µL depletion condition than the 16 µL depletion condition ($\beta_{8µL-16µL}$ = 1.589 trials, SE = 0.155, t(7.000) = 10.28, p < .001).

These first two experiments confirm that rats qualitatively follow the predictions of MVT, but consistently overharvest. There are many possible explanations for this pattern of overharvesting, including an aversion to leaving the offer of reward within a patch and nonlinear reward utility (*Wikenheiser et al., 2013*; *Carter and Redish, 2016*; *Constantino and Daw, 2015*). The Scale Experiment was conducted in an effort to distinguish between these hypotheses by manipulating the scale of time delays and rewards. Long-term rate maximization predicts that an increase in reward size in proportion to reward delay should have no effect on the number of harvests per patch, as the reward rate across trials would be equal. But if animals' perception of reward or time is nonlinear, a manipulation of scale will affect their subjective point of equality and predict a change in behavior across the two environments. The scale of rewards and delays was manipulated in the following manner: patches started with (A) 90 or (B) 180 µL of reward, depleted at a rate of (A) 8 or (B) 16 µL/trial, and the duration of harvest trials and travel time between patches was (A) 10 or (B) 20 s. Rats visited a range of 60–212 patches per condition. They overharvested in both A and B conditions ($\beta_{rat-MVT}$ = 4.374 trials, SE = 0.153, t(6.900) = 28.597, p < .001) and, contrary to predictions of MVT, they stayed in patches significantly longer and overharvested to a greater degree in the B condition that provided larger rewards but at proportionately longer delays ($\beta$ = 1.937 trials, SE = 0.193, t(6.972) = 9.996, p < .001; *Figure 1C*). This finding suggests that a nonlinearity in the perception of reward value and/or time contributes to overharvesting.

To distinguish between biases in perception of reward, such as nonlinear reward utility, and time, such as temporal discounting or insensitivity to post-reward delays, the Pre-vs-Post Experiment

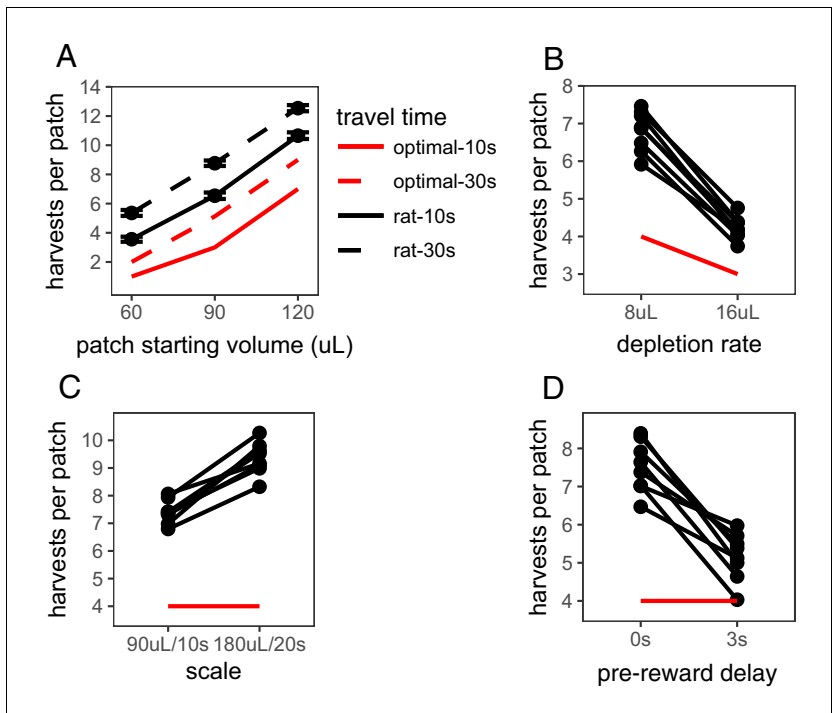

**Figure 1.** Rat foraging behavior. Rat foraging behavior in the (**A**) Travel Time, (**B**) Depletion Rate, (**C**) Scale, and (**D**) Pre-vs-Post Experiments. In (**A**), points and error bars represent mean ± standard error. In (**B-D**), points and connecting lines represent behavior of each individual rat. Red lines indicate optimal behavior (per MVT).
DOI: https://doi.org/10.7554/eLife.48429.003

The following source data and figure supplement are available for figure 1:

**Source data 1.** Trial-by-trial foraging behavior.
DOI: https://doi.org/10.7554/eLife.48429.005
**Figure supplement 1.** Diagram of the foraging task.
DOI: https://doi.org/10.7554/eLife.48429.004

directly tested rats sensitivity to time delays before vs. after reward. In this experiment, in one condition, rats received reward immediately after lever press followed by a post-reward delay of ~13 s before the start of the next trial. In the other condition, there was a 3 s pre-reward delay between lever press and receiving reward followed by a shorter post-reward delay of ~10 s. The total time of each trial was held constant between conditions (15 s total), so there was no difference in reward rates. Both MVT and nonlinear reward utility predict that the placement of delays is inconsequential and that rats will behave similarly in both conditions. Both temporal discounting and insensitivity to post-reward delays predict that rats will value the immediate reward more than the delayed reward and thus, would leave patches earlier in the condition with the pre-reward delay. Consistent with predictions of temporal discounting and insensitivity to post-reward delays, and contrary to predictions of MVT and nonlinear reward utility, rats left patches earlier in the environment with the pre-reward delay ($\beta$ = 2.345 trials, SE = 0.313, t(7.017) = 7.503, p < .001; *Figure 1D*). This result suggests that a bias in rats' perception of time or the way in which they perceive delayed reward values contributes to overharvesting.

To determine whether the preference for immediate rewards can be explained by insensitivity to post-reward delays, a fifth foraging experiment, the 'Post-Reward Delay Experiment,' directly tested rats' sensitivity to post-reward delays. A separate cohort of rats (n = 8) was used for this experiment. Rats were tested in two conditions in this experiment: a short (3 s) or long (12 s) post-reward delay. The total time of harvest trials was not held constant; the longer post-reward delay increased the time to harvest from the patch. Since the longer post-reward delay increases the cost of harvesting from the patch relative to the cost of traveling to a new patch, MVT predicts that rats should leave patches earlier. Prior studies of intertemporal choice behavior have shown that animals are

insensitive to post-reward delays, suggesting that they are only concerned with maximizing short-term reward rate (*Stephens, 2001*; *Bateson and Kacelnik, 1996*) or that they may not have learned the duration of post-reward delays, and underestimate this duration in their decision process (*Pearson et al., 2010*; *Blanchard et al., 2013*). Consistent with MVT, rats were sensitive to the post-reward delay, leaving patches earlier in the 12 s delay condition ($\beta$ = 1.411 trials, SE = 0.254, t(6.966) = 5.546, p < .001; *Figure 2A*). If rats were sensitive to the delay, but underestimated its duration, one would still expect rats to overharvest to a greater degree due to the longer delay. There was no difference in the degree to which rats overharvested between the 3 s and 12 s delay conditions ($\beta_{rat-MVT;3\ s-12\ s}$ = 0.340 trials, SE = 0.286, t(6.963) = 1.188, p = 0.274). This finding suggests that overharvesting in this experiment is not due to insensitivity to post-reward delays. However, it is possible that this finding could be explained by other forms of altered time perception that remain to be described.

The data from the foraging experiments described above suggest that rats exhibit time preferences in the foraging task. In a final 'Intertemporal Choice Experiment,' we tested whether the same rats that participated in the Post-Reward Delay Experiment would exhibit similar time preferences in a standard intertemporal choice (i.e. a delay-discounting) task. This task consisted of a series of 20-trial episodes. On each trial, rats pressed either the left or right lever to receive a smaller-sooner (SS) reward of 40 µL after a 1 s delay or a larger-later (LL) reward of 40, 80, or 120 µL after a 1, 2, 4, or 6 s delay. For the first 10 trials of each episode, rats were forced to press either the left or right lever to learn the value and delay associated with that lever (only one lever extended on each of these trials). For the last 10 trials of an episode, both levers extended and rats were free to choose. The LL reward value and delay, and the LL lever (left or right) were randomly selected at the start of each episode. Rats were tested in two different versions of this task: one in which the post-reward delay was held constant, such that the longer pre-reward delays reduced reward rate (constant delay); and another in which the time of the trial was held constant, such that longer pre-reward delays resulted in shorter post-reward delays to keep reward rate constant (constant rate). MVT, which maximizes long-term reward rate, predicts that rats would be sensitive to the pre-reward delay in the constant delay condition but not the constant trial condition (in which the pre-reward delay does not affect reward rate).

Rats were given three training sessions to learn the structure of the intertemporal choice task after previously being tested in the foraging task, then they were tested for an additional 13 sessions

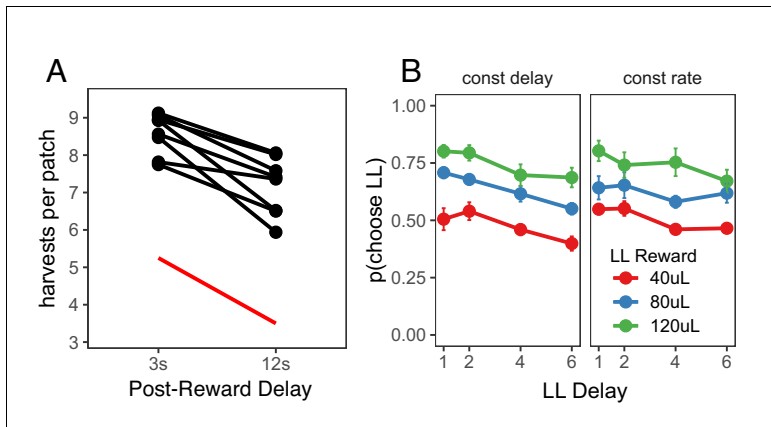

**Figure 2.** Post-reward delay foraging and intertemporal choice behavior. (A) Rat behavior in the Post-Reward Delay Experiment. Points and lines represent behavior of individual rats. Red line indicates optimal behavior (per MVT). (B) Rat behavior in the intertemporal choice task. Points and error bars represent mean ± standard error for each condition.

DOI: https://doi.org/10.7554/eLife.48429.007

The following source data is available for figure 2:

**Source data 1.** Trial-by-trial intertemporal choice behavior.

DOI: https://doi.org/10.7554/eLife.48429.008

in each condition, participating in a range of 590–2810 free choice trials per condition (constant delay vs. constant rate). Each free choice trial within each episode was counted as a separate observation. Choice data were analyzed using a generalized linear mixed-effects model (i.e. a mixed-effects logistic regression) to examine the effect of the size of the LL reward, the length of the LL delay, task condition (constant delay vs. constant rate), and their interactions on decisions to choose the LL vs. SS option, with random intercepts and random slopes for the effects of LL reward, LL delay, and task condition for each rat. Three post-hoc comparisons were used to test the effects of (i) LL reward and (ii) LL delay within each condition, and (iii) LL delay between the constant delay and constant rate conditions (*Figure 2B*). (i) In both conditions, rats were more likely to choose larger LL rewards (constant delay: β = 0.477, SE = 0.090, $\chi^2(1)$=28.320, p < .001; constant rate: β = 0.450, SE = 0.089, $\chi^2(1)$ = 25.378, p < .001), showing that they were sensitive to reward magnitude. (ii) They were also sensitive to the pre-reward delay in both conditions (constant delay: β = −0.240, SE = 0.023, $\chi^2(1)$ = 104.882, p < .001; constant rate: β = −0.152, SE = 0.022, $\chi^2(1)$ = 46.919, p < .001). On average, rats were equally likely to select the LL option across conditions — the main effect of task condition was not significant (β=0.010, SE = 0.105, z = 0.092, p = 0.927). (iii) However, rats were less sensitive to increasing pre-reward delays when pre-reward delays did not affect reward rate (in the constant rate condition), indicated by a change in LL delay slope between conditions (β = 0.088, SE = 0.026, $\chi^2(1)$ = 11.376, p < .001). Overall, rats exhibited similar time preferences in the foraging and intertemporal choice tasks: they valued rewards less with longer delays until receipt but they were sensitive to opportunity costs (e.g. time delays between receiving reward and future decisions).

## Quasi-hyperbolic discounting best explains behavior across all tasks

To test whether a common set of cognitive biases could explain time preferences in both the foraging and intertemporal choice tasks, both tasks were modeled as continuous time semi-markov processes. These models consisted of a set of states that represented the time between each event in each of the tasks (e.g. cues turning on/off, lever press, reward delivery; for state space diagrams of both tasks, see *Figure 3—figure supplement 1* and *Figure 4—figure supplement 1*). These models assumed that animals have learned the appropriate structure of the task (i.e. the time spent and reward obtained in each state) unless otherwise noted. The value of a given state was the discounted value of all future rewards available from that state, and the agent chose the option that yielded the greatest discounted future reward via a stochastic process. As the discount factor approached 1 (i.e. no temporal discounting), this model converged to long-term reward maximization, equivalent to MVT. Additional parameters were added to the model to test four specific hypotheses for suboptimal foraging behavior: (i) subjective costs associated with leaving a patch, in which the value of leaving was reduced by a 'cost' term; (ii) nonlinear reward utility, in which the subjective utility of a reward increased sublinearly with respect to the reward magnitude; (iii) biased time perception, which assumed that animals underestimate post-reward delays, possibly due to insufficient learning of task structure (*Blanchard et al., 2013*; *Pearson et al., 2010*), or overestimate pre-reward delays; and (iv) temporal discounting. A brief description of each hypothesis and its general predictions can be found in *Table 2*. For each model, group level parameters and parameters for each individual rat were fit simultaneously using an expectation-maximization algorithm (*Huys et al., 2011*). Parameters were fit to each experiment separately (one set of parameters for both conditions in each experiment). Model predictions were calculated separately for each rat, using the rat's individual parameters. Full details for all models, fitting procedures, and model comparison can be found in the Materials and methods.

Subjective costs to leave a patch and nonlinear reward utility have explained suboptimal foraging behavior in prior studies that have manipulated opportunity costs (e.g. travel time or pre-reward delays) and depletion rate (*Constantino and Daw, 2015*; *Wikenheiser et al., 2013*; *Carter and Redish, 2016*). However, these factors are insensitive to the placement of time delays (pre- vs. post-reward) and thus, cannot explain the preference for more immediate rewards. Consistent with these prior studies, the subjective costs and nonlinear reward utility models explained overharvesting in the Travel Time, Depletion Rate, and Post-Reward Delay Experiments, but they failed to explain time preferences in the Pre-vs-Post foraging experiment (*Figure 3—figure supplement 2*).

**Table 2.** Description of the hypotheses for overharvesting with general, qualitative predictions for the degree of overharvesting in each experiment.

Quantitative predictions depend on the exact formalization of each model and its specific parameters.

| | Hypothesis | Experimental predictions |
|---|---|---|
| Subjective Costs | A cost term $c$ reduces the value of leaving a patch. Predicts greater overharvesting with higher $c$. Not affected by specific manipulations to reward or time. | Rats will follow qualitative predictions of the Marginal Value Theorem, but exhibit an equal degree of overharvesting across conditions in each experiment. |
| Nonlinear Reward Utility | Subjective value increases sublinear to reward magnitude. Predicts greater overharvesting with steeper utility functions with larger rewards. | Rats will exhibit an equal degree of ovarharvesting in all experiments except for the Scale experiment. In the Scale experiment, rats will overharvest more in the conditions with larger rewards. |
| Biased Time Perception | i) Post-reward delays perceived as shorter, ii) pre-reward delays perceived as longer, or iii) longer delays (irrespective of their placement) perceived as shorter. All three hypotheses predict greater overharvesting with longer delays. | Rats will exhibit a greater degree of overharvesting in the condition with longer delays in the Scale environment, in the condition with the longer post-reward delay in the Pre-vs-Post experiment, and in the condition with longer post-reward delay in the Post-Reward Delay experiment |
| Temporal Discounting | Value of future rewards discounted due to delay to receive them. Predicts greater overharvesting with greater levels of discounting and with longer delays | Rats will overharvest to a greater degree in the conditions with longer delays in the Scale and Post-Reward Delay experiments and they will leave patches earlier due to the longer pre-reward delay in the Pre-vs-Post experiment. |

DOI: https://doi.org/10.7554/eLife.48429.009

We next examined whether biased time perception and temporal discounting could explain suboptimal foraging behavior across all tasks. Three implementations for biased time perception were tested: linear underestimation of post-reward delays ($postDelay = \alpha * postDelay$), non-linear underestimation of post-reward delays ($postDelay = postDelay^\alpha$), and overestimation of pre-reward delays ($preDelay = \alpha * preDelay$). For temporal discounting, we tested the two common single-parameter discounting functions, exponential ($value = e^{-\beta * time} * reward$) and standard hyperbolic ($value = reward/(1 + k * time)$), and two more flexible discounting models: constant sensitivity discounting (*Ebert and Prelec, 2007*; *Zauberman et al., 2009*), which predicts hyperbolic time preferences due to insensitivity to longer delays ($value = e^{-\beta * time^\alpha} * reward$); and quasi-hyperbolic discounting, formalized as two competing exponential discounting systems ($value = [\omega * e^{-\beta * time} + (1 - \omega) * e^{-\delta * time}] * reward$; (*Laibson, 1997*; *McClure et al., 2007*). All of these models qualitatively predicted rat behavior across foraging experiments (*Figure 3*, *Figure 3—figure supplement 3*, *Figure 3—figure supplement 4*).

To determine which model provided the best quantitative fit, we compared the group-level Bayesian Information Criterion (integrated BIC or iBIC; *Huys et al., 2011*; *Huys et al., 2012*) of all models in each of the foraging tasks. To compare across tasks, we took the sum of iBIC for each model. The quasi-hyperbolic discounting model had the lowest sum of iBIC, the constant sensitivity discounting model the second lowest, and the hyperbolic discounting model third (*Figure 3*). These three models were also among the lowest iBIC values for each individual experiment *Figure 3—figure supplement 5*). All three of these models predict that animals will exhibit hyperbolic time preferences, suggesting that suboptimal foraging behavior observed in these experiments is due to time preferences.

Next, we tested whether the quasi-hyperbolic discounting model that provided the best fit to foraging behavior could also explain behavior in the intertemporal choice task. As in the model of the foraging task, the model of the intertemporal choice task took into account all future rewards, including rewards from future episodes (see abbreviated state space diagram in *Figure 4—figure supplement 1*). We tested the nonlinear reward utility, biased time perception and temporal discounting models in this task (the subjective cost model does not apply to this task). Again, the quasi-hyperbolic discounting model had the lowest iBIC and hyperbolic discounting model the second lowest, but the constant sensitivity model had a higher iBIC than the biased time perception models (*Figure 4*). As the constant sensitivity model produces hyperbolic time preferences via insensitivity to longer delays, these results suggest that hyperbolic time preferences without insensitivity to delays is the best explanation for rat intertemporal choice behavior.

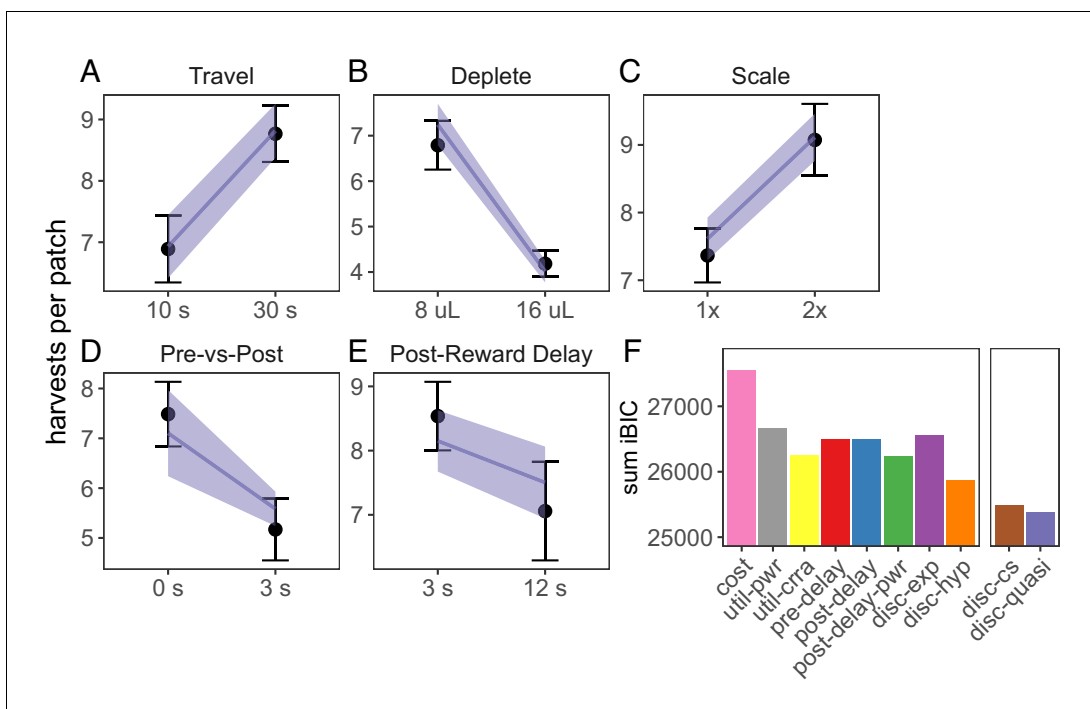

**Figure 3.** Model predictions for foraging tasks. (A-E) Predictions of the best fit quasi-hyperbolic discounting model to all foraging tasks. Points and error bars represent mean ± standard deviation of the means for each individual rat; lines and ribbon represent the mean ± standard deviation of the means of the model-predicted behavior for each individual rat. (F) The sum of iBIC scores across all foraging tasks for each model. Cost = subjective cost model, util-pwr and util-crra = nonlinear reward utility with power and CRRA function respectively, pre-del = linear overestimation of pre-reward delays, post-del = linear underestimation of post-reward delays, post-del-pwr=underestimation of post-reward delays according to a power function, disc-exp = exponential discounting, disc-hyp = hyperbolic discounting, disc-cs = constant sensitivity discounting, disc-quasi = quasi hyperbolic discounting.

DOI: https://doi.org/10.7554/eLife.48429.010

The following figure supplements are available for figure 3:

**Figure supplement 1.** State space diagram of the foraging task.

DOI: https://doi.org/10.7554/eLife.48429.011

**Figure supplement 2.** Predictions of the best fit subjective cost and nonlinear reward utility models.

DOI: https://doi.org/10.7554/eLife.48429.012

**Figure supplement 3.** Predictions of the best fit biased time perception models.

DOI: https://doi.org/10.7554/eLife.48429.013

**Figure supplement 4.** Predictions of the best fit discounting models.

DOI: https://doi.org/10.7554/eLife.48429.014

**Figure supplement 5.** iBIC for each model for each foraging experiment.

DOI: https://doi.org/10.7554/eLife.48429.015

If hyperbolic time preferences reflect a common explanation for suboptimal decision-making, then it might be expected that a model of behavior fit to one task could predict a rat's behavior in the other task. To test the external validity of this hypothesis, data from each task were separated into three subsets. The best fitting model from both tasks, the quasi-hyperbolic discounting model, was fit to two subsets of data from one task, then the negative log likelihood (-LL) of the data was assessed on the left out sample from both tasks. This process was repeated such that each subset served as the left out sample. To determine which discount function provided the better fit to data from each task, we calculated the difference in -LL of the left out sample between the model fit to intertemporal choice data and the model fit to foraging data (-LL difference = $-LL_{itc}$ - $LL_{forage}$). Since smaller -LL indicates a better fit, a positive -LL difference indicates that the discount function fit to

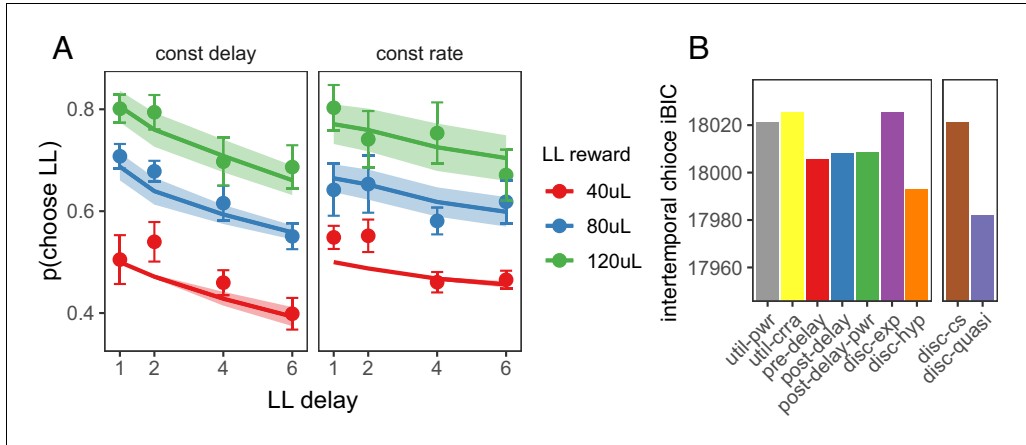

**Figure 4.** Model predictions for the intertemporal choice task. (**A**) Quasi-hyperbolic model predictions for the intertemporal choice task. Points and error bars represent the mean ± standard error of individual rat behavior; lines and ribbon represent mean ± standard error of model predicted behavior for each individual rat. (**B**) The iBIC score for each model for the delay discounting experiment. Util-pwr and util-crra = nonlinear reward utility with power and CRRA function respectively, pre-del = linear overestimation of pre-reward delays, post-del = linear underestimation of post-reward delays, post-del-pwr=underestimation of post-reward delays according to a power function, disc-exp = exponential discounting, disc-hyp = hyperbolic discounting, disc-cs = constant sensitivity discounting, disc-quasi = quasi hyperbolic discounting.

DOI: https://doi.org/10.7554/eLife.48429.016

The following figure supplements are available for figure 4:

**Figure supplement 1.** State space diagram of the intertemporal choice task.

DOI: https://doi.org/10.7554/eLife.48429.017

**Figure supplement 2.** Comparison of all-future horizon and one-trial horizon discounting models.

DOI: https://doi.org/10.7554/eLife.48429.018

---

foraging data provided a better fit (i.e. the foraging -LL was lower than the intertemporal choice -LL). For the foraging task, discounting functions fit to foraging data provided a better fit than discounting functions fit to intertemporal choice data for all eight rats. Interestingly, for the intertemporal choice task, discounting functions fit to foraging data provided a better fit than discounting functions fit to intertemporal choice data for 3 of 8 rats (*Figure 5*). The quasi-hyperbolic model fit to the foraging task generalized well to the intertemporal choice task, providing support for the idea that foraging and intertemporal choice can be described by a common discount function.

With temporal discounting models that consider all future rewards, the more flexible quasi-hyperbolic discounting function provided the best fit to behavior across tasks. We next directly tested whether considering future rewards affects the fit of discounting models to intertemporal choice data and the estimates of discount factors compared to temporal discounting models that only consider the next reward (one-trial horizon models). We fit one-trial horizon models for all of the previously tested discounting functions — exponential, hyperbolic, constant sensitivity, and quasi-hyperbolic discounting — and compared them to the discounting models that considered all future rewards (all-future horizon models). For all discounting functions, the all-future horizon models had lower iBIC than one-trial horizon models (*Figure 4—figure supplement 2*). To compare the discount factors of each model (for the quasi-hyperbolic function that has two discount factors, we used the slow discounting β), we performed paired t-tests between log transformed discount factors measured by the all-future horizon models vs. discount factors measured by the one-trial horizon models. Measured discount factors were lower for the all-future models for all discounting functions (exponential: t(7) = 22.439, p < .001; hyperbolic: t(7) = 7.000, p < .001; constant sensitivity: t(7) = 15.497, p < .001; quasi-hyperbolic: t(7) = 25.322, p < .001; p-values adjusted using Bonferroni correction). Lastly, we tested whether the all-future horizon quasi-hyperbolic discounting model fit to the intertemporal choice data would predict foraging behavior better than the one-trial horizon quasi-hyperbolic discounting model fit to intertemporal choice data. For 6 of 8 rats, parameters fit to the one-trial horizon model produced a better fit to foraging behavior than parameters fit to the all-future

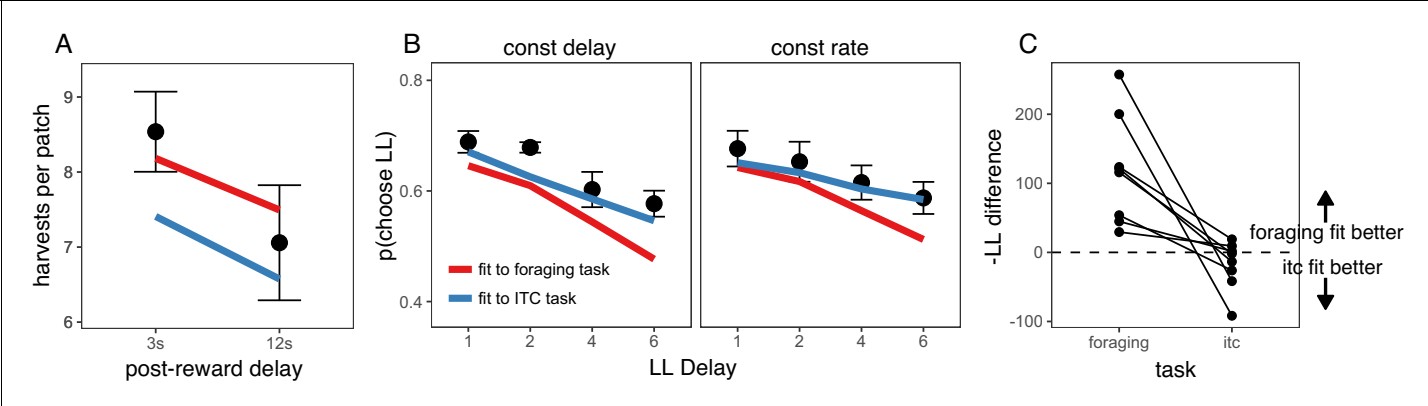

**Figure 5.** Cross-task model predictions. (A) Predicted foraging behavior for quasi-hyperbolic model parameters fit to either the foraging task (red line) or delay discounting task (DD; blue line). Black points and error bars represent mean ± standard error of rat data. (B) Predicted intertemporal choice behavior for quasi-hyperbolic model parameters fit to data from either the foraging or delay discounting task, plotted against rat behavior. (C) The difference in negative log likelihood of the left out sample of foraging data (left) or intertemporal choice data (right) between parameters fit to the intertemporal choice task and parameters fit to the foraging task. A negative -LL difference indicates the negative log likelihood of the data for parameters fit to the intertemporal choice task was lower than for parameters fit to the foraging task. Each point and line represents data from individual rats.

DOI: https://doi.org/10.7554/eLife.48429.019

horizon model. Overall, using full horizon temporal discounting models explained more of the intertemporal choice data, produced smaller estimates of discounting factors, but in the present study, it did not improve the ability of a model fit to intertemporal choice data to predict foraging behavior.

## Discussion

In foraging studies, animals exhibit behavior that conforms qualitatively to predictions made by optimal foraging theory (i.e., the MVT), choosing to leave a patch when its value falls below that of the average expected value of other(s) available in the environment. However, an almost ubiquitous finding is that they overharvest, leaving a patch when its value falls to a value lower than the one predicted by MVT. Given that the rewards available within the current patch are generally available sooner than those at other patches due to travel time, one interpretation of overharvesting is that this reflects a similarly prevalent bias observed in intertemporal choice tasks, in which animals consistently show a greater preference for smaller more immediate rewards over later delayed rewards than would be predicted by optimal (i.e,. exponential) discounting of future values. However, in prior studies, models of intertemporal choice behavior have been poor predictors of foraging behavior (*Blanchard and Hayden, 2015*; *Carter and Redish, 2016*). Here, we show that in a carefully designed series of experiments, rats exhibit similar time preferences in foraging and intertemporal choice tasks, and that a quasi-hyperbolic discounting model can explain the rich pattern of behaviors observed in both tasks.

The foraging behavior we observed was consistent with previous studies of foraging behavior in rats, monkeys, and humans, while also revealing novel aspects of overharvesting behavior. Consistent with prior studies, rats stayed longer in patches that yielded greater rewards, stayed longer in all patch types when the cost of traveling to a new patch was greater, left patches earlier when rewards depleted more quickly, and consistently overharvested (*Constantino and Daw, 2015*; *Hayden et al., 2011*; *Kane et al., 2017*). Our experiments also demonstrated that in certain environments rats violate qualitative predictions of MVT. Rats overharvested more when reward amount and delay were increased, even though reward rate was held constant, and they were differentially sensitive to whether the delay was before the receipt of the proximal reward or following its delivery. These findings supported the conjecture that overharvesting is related to time preferences.

A number of studies have found that the preference for smaller, more immediate rewards can be explained by insensitivity to post-reward delays (*Bateson and Kacelnik, 1996*; *Blanchard et al., 2013*; *Pearson et al., 2010*; *Stephens, 2001*; *Mazur, 1991*). One hypothesis for why animals fail to

incorporate post-reward delays into decisions is that they haven't learned the structure of the task well, and thus cannot accurately predict future post-reward delays. Accordingly, providing explicit cues for the post-reward delays or increasing the salience of post-reward delays helps animals incorporate these delays into their decisions, reducing the bias towards selecting smaller, more immediate rewards over larger, delayed ones (*Pearson et al., 2010*; *Blanchard et al., 2013*). However, in the present study, rats were sensitive to post-reward delays in both the foraging and intertemporal choice task, providing further evidence that the preference for smaller, more immediate rewards in both tasks is due to time preferences and not a poor understanding of the task structure. Furthermore, quantitative modeling supported the hypothesis that suboptimal behavior was driven by time preferences rather than insensitivity to delays.

The idea that animals exhibit similar decision biases in foraging and intertemporal choice paradigms, and that these biases can be explained by a common model of discounting, is in conflict with prior studies that found that animals are better at maximizing long-term reward rate in foraging than in intertemporal choice tasks, and that delay discounting models of intertemporal choice tasks are poor predictors of foraging behavior (*Stephens, 2008*; *Blanchard and Hayden, 2015*; *Carter et al., 2015*; *Carter and Redish, 2016*). It has been argued that animals may perform better in foraging tasks because decision-making systems have evolved to solve foraging problems rather than two-alternative intertemporal choice problems (*Blanchard and Hayden, 2015*; *Stephens et al., 2004*; *Stephens, 2008*). This idea has been challenged by a recent study of human decision-making in foraging and intertemporal choice tasks, finding that a long-term rate maximization model explained both foraging and intertemporal choice behavior better than a standard hyperbolic discounting model (*Seinstra et al., 2018*). Results from the present study support the interpretation that foraging and intertemporal choice behavior can be explained via a common model, but suggest that this model is quasi-hyperbolic discounting. We found that a quasi-hyperbolic discounting model provided the best explanation to rat behavior across multiple foraging tasks and an intertemporal choice task, and that a quasi-hyperbolic discounting model fit to individual rat foraging behavior can predict their intertemporal choice behavior.

Two potential explanations for why temporal discounting models have failed to predict foraging behavior in prior studies are that (i) prior studies have only tested single-parameter exponential and hyperbolic discounting functions, whereas the present study also tested the more flexible quasi-hyperbolic discounting function; and (ii) in most of these studies, models of intertemporal choice tasks have only considered the most proximal reward (the reward received as a consequence of the decision at hand). This assumption seems appropriate as, in most intertemporal choice tasks, opportunities for future rewards do not depend on the current decision, so the value of rewards received for future decisions are equal for both the SS and LL rewards. But in foraging tasks, future opportunities for reward depend on current decisions, so it is critical for foraging models to include all future rewards into estimates of reward value. For this reason, comparing discount functions fit to intertemporal choice models that consider all future reward may provide better estimates of foraging behavior than discount functions fit to intertemporal choice models that only consider rewards from the most proximal decision. Consistent with this hypothesis, we found that adding the value of future rewards to intertemporal choice models reduces estimates of discount factors. However, with our data, the quasi-hyperbolic discounting model fit to the intertemporal choice task that included all future rewards did not predict foraging behavior better than an equivalent model that only considered the most proximal reward. One reason why including all future rewards may not have improved cross-task predictions is that, in the present study, the quasi-hyperbolic discounting model fit to the intertemporal choice task predicted less overharvesting than was exhibited by rats in the foraging task. Reducing estimates of the discount factor with a model that considers all future rewards predicts even less overharvesting (i.e. behavior that is closer to long-term reward maximization). But in other studies, temporal discounting models typically predict greater overharvesting than is exhibited by animals (*Blanchard and Hayden, 2015*; *Carter et al., 2015*). In these cases, obtaining smaller, potentially more accurate estimates of discount factors by including all future rewards into intertemporal choice models may improve cross-task predictions.

Although quasi-hyperbolic discounting provided the best singular explanation for rat behavior across our tasks, many of the other models tested were capable of explaining some of the biases exhibited by rats. Thus, we cannot exclude the possibility that subjective costs, diminishing marginal utility, and/or biased estimation of time intervals may independently contribute to suboptimal

decision-making. Furthermore, additional hypotheses or additional variants of the above-mentioned hypotheses that have not been tested in the present study may provide alternative explanations for suboptimal decision making in foraging and intertemporal choice tasks. Importantly, our data indicate that quasi-hyperbolic discounting may provide a link between foraging and intertemporal choice tasks, and it highlights the importance of future work considering the source of time preferences. These observations are buttressed by recent theoretical work demonstrating that the appearance of time preferences in intertemporal choice tasks can emerge rationally from a value construction process by which estimates increase in variability with the delay until reward receipt — an account that shares features with the short-term rate maximization hypotheses (*Stephens et al., 2004*). Under this account, 'as-if' discounting is hyperbolic when variability increases linearly with delay (*Gabaix and Laibson, 2017*). Further, a sequential sampling model of two-alternative forced choice (*Bogacz et al., 2006*), parameterized such that outcome delay scales variability in this way, has recently been shown to capture key dynamical features of both patch foraging (*Davidson and El Hady, 2019*) and hyperbolic discounting in intertemporal choice (*Hunter et al., 2018*). Future work should build on these findings to explore directly whether the common biases identified here reflect a core computation underlying sampling and decision-making under uncertainty and across time.

## Materials and methods

### Animals

Adult Long-Evans rats were used (Charles River, Kingston, NY). One group of eight rats participated in the scale, travel time, depletion rate, and handling time experiments (in that order), a different set of eight rats were tested on the post-reward delay foraging experiment then the delay discounting task. Rats were housed on a reverse 12 hr/12 hr light/dark cycle. All behavioral testing was conducted during the dark period. Rats were food restricted to maintain a weight of 85–90% ad-lib feeding weight, and were given ad-lib access to water. All procedures were approved by the Princeton University and Rutgers University Institutional Animal Care and Use Committee.

### Foraging task

Animals were trained and tested as in *Kane et al. (2017)*. Rats were first trained to lever press for 10% sucrose water on an FR1 reinforcement schedule. Once exhibiting 100+ lever presses in a one hour session, rats were trained on a sudden patch depletion paradigm — the lever stopped yielding reward after 4–12 lever presses — and rats learned to nose poke to reset the lever. Next rats were tested on the full foraging task.

A diagram of the foraging task is in *Figure 1—figure supplement 1*. On a series of trials, rats had to repeatedly decide to lever press to harvest reward from the patch or to nose poke to travel to a new, full patch, incurring the cost of a time delay. At the start of each trial, a cue light above the lever and inside the nose poke turned on, indicating rats could now make a decision. The time from cues turning on until rats pressed a lever or nose poked was recorded as the decision time (DT). A decision to harvest from the patch (lever press) yielded reward after a short pre-reward delay (referred to as the handling time delay, simulating the time to 'handle' prey after deciding to harvest). Reward (sucrose water) was delivered when the rat entered the reward magazine. The next trial began after an inter-trial interval (ITI). To control the reward rate within the patch, the length of the ITI was adjusted based on the DT of the current trial, such that the length of all harvest trials was equivalent. With each consecutive harvest, the rat received a smaller volume of reward to simulate depletion from the patch. A nose poke to leave the patch caused the lever to retract for a delay period simulating the time to travel to a new patch. After the delay, the opposite lever extended, and rats could harvest from a new, replenished patch.

Details of the foraging environment for each experiment can be found in *Table 1*. For each experiment, rats were trained on a specific condition for 5 days, then tested for 5 days. Conditions within experiments were counterbalanced.

### Foraging data analysis

Rat foraging behavior was assessed using linear mixed effects models. Models were fit using the lme4 package in R (*Bates et al., 2015*). The lme4 package provides only t-statistics for fixed effects;

p-values were calculated using the lmerTest package (*Kuznetsova et al., 2017*), which uses Scatterwaithe's method to approximate the degrees of freedom for the t-test. In the Travel Time Experiment, we assessed the effect of starting volume of the patch and the travel time on number of harvests per patch, with random intercepts and random slopes for both variables across subjects (lme4 formula: $HarvestsPerPatch \sim PatchStartingVolume * TravelTime + (PatchStartingVolume + TravelTime \,||\, Rat)$). In all other foraging experiments, we assessed the effect of experimental condition on harvests per patch, with random intercepts and random effect of experimental condition across subjects (lme4 formula: $HarvestsPerPatch \sim Condition + (Condition \,|\, Rat)$).

We also tested whether rats overharvested relative to MVT predictions in each experiment, and whether the degree of overharvesting was different between conditions within each experiment. To do so, we subtracted the MVT predicted number of harvests in each patch from the observed number of harvests (see 'Foraging Models' section for details on the calculation of the optimal number of harvests). Mixed effects models were used to fit an intercept and effect of experimental condition on the difference from optimal number of harvests (lme4 formula: $DifferenceFromOptimal \sim Condition + (Condition \,|\, Rat)$). In this model, an intercept greater than zeros would indicate that rats harvested more trials than was predicted by MVT, and a difference in the effect of task condition would indicate that the degree to which rats differed from optimal was affected by the task condition.

## Intertemporal choice task

Rats were immediately transferred from the foraging task to the intertemporal choice task with no special training; rats were given three 2 hr sessions to learn the structure of the new task. This task consisted of a series of episodes that lasted 20 trials. At the beginning of each episode one lever was randomly selected as the shorter-sooner lever, yielding 40 μL of reward following a 1 s delay. The other lever (larger-later lever) was initialized to yield a reward of 40, 80, or 120 μL after a 1, 2, 4 or 6 s delay. For the first 10 trials of each episode, only one lever extended, and rats were forced to press that lever to learn its associated reward value and delay. The last four forced trials (trials 7–10) were counterbalanced to reduce the possibility of rats developing a perseveration bias. For the remaining 10 trials of each episode, both levers extended, and rats were free to choose the option they prefer. At the beginning of each trial, cue lights turned on above the lever indicating rats could now make a decision. Once the rat pressed the lever, the cue light turned off, and the delay period was initiated. A cue light turned on in the reward magazine at the end of the delay period, and rats received reward as soon as they entered the reward magazine. Reward magnitude was cued by light and tone. Following reward delivery, there was an ITI before the start of the next trial. At the completion of the episode, the levers retracted, and rats had to nose poke to begin the next episode, which reset the larger-later reward and delay.

## Intertemporal choice data analysis

Intertemporal choice data was analyzed using a mixed effects logistic regression, examining the effect of larger-later reward value, larger-later delay, and task condition on rats choices, with random intercepts and random effects for all three variables. This model was fit as a generalized linear mixed effects model using the lme4 package in R (lme4 formula: $ProbLL \sim RewardLL * DelayLL * Condition + (RewardLL + DelayLL + Condition \,||\, Rat)$; *Bates et al., 2015*). Post-hoc comparisons of interest were tested using the phia package in R (*De Rosario-Martinez, 2015*), using Holm's method to correct for multiple comparisons.

## Foraging models

All models were constructed as continuous time semi-markov processes. This provided a convenient way to capture the dynamics of timing in both tasks, such as slow delivery and consumption of reward (up to 6 s for the largest rewards). To model the foraging task, each event within the task (e.g. cues turning on/off, lever press, reward delivery, etc.) marked a state transition (abbreviated state space diagram in *Figure 3—figure supplement 1*. All state transitions were deterministic, except for decisions to stay in vs. leave the patch, which occurred in 'decision' states (the time between cues turning on at the start of the trial and rats performing a lever press or nosepoke). In decision states, a decision to stay in the patch transitioned to the handling time state, then reward

state, ITI state, and to the decision state on the next trial. A decision to leave transitioned to the travel time state, then to the first decision state in the patch. Using the notation of *Bradtke and Duff, 1995*, the value of staying in state $s$, $Q(stay, s)$, is the reward provided for staying in state $s$, $R(stay, s)$, plus the discounted value of the next state:

$$Q(stay, s) = R(stay, s) + \gamma(stay, s) * V(s_{next})$$

where $\gamma(stay, s)$ is the discount applied to the value of the next state for staying in state $s$, and $V(s_{next})$ is the value of the next state in the patch. For all non-decision states, rats did not have the option to leave the patch, so for these states, $V(s) = Q(stay, s)$. For decision states, the value of the state was the greater of $Q(stay, s)$ and $Q(leave)$.

For simplicity, we assume the time spent in a given state is constant, calculated as the average amount of time a given rat spent in the state. Under this assumption, the reward in a given state, $R(stay, s)$, is equal to the reward rate provided over the course of the state, $r(s)$, multiplied by the time spent in that state $T(s)$, discounted according to discount factor $\beta$:

$$R(stay, s) = \frac{1 - e^{-\beta * T(s)}}{\beta} * r(s), \text{and}$$

$$\gamma(stay, s) = e^{-\beta * T(s)}.$$

The value of leaving a patch, $Q(leave)$, was equal to the discounted value of the first state in the next patch, $V(s_{first})$:

$$Q(leave) = \gamma(leave) * V(s_{first})$$

where $\gamma(leave)$ is the discount factor applied to the next state in the first patch. Assuming no variance in the travel time $\tau$, $\gamma(leave) = e^{-\beta * \tau}$. Per MVT, we assumed rats left patches at the first state in the patch in which $Q(stay, s) \leq Q(leave)$. To model variability in the trial at which rats left patches, we added gaussian noise to $Q(leave)$. As decisions within each patch are not independent, the patch leaving threshold did not vary trial-by-trial, but rather patch by patch, such that the cumulative probability that a rat has left the patch by state $s$, $\pi(leave, s)$, was the probability that $Q(stay, s) \leq Q(leave) + Q(leave) * \epsilon$, where $\epsilon \sim \mathcal{N}(0, \sigma^2)$, with free parameter $\sigma$. $\epsilon$ scaled with $Q(leave)$ to enable comparisons across conditions within experiments.

The optimal policy for a given set of parameters was found using value iteration (*Sutton and Barto, 1998*). MVT predictions (maximization of undiscounted long-term reward rate) were determined by fixing the discount factor $\beta = .001$ and assuming no decision noise ($\epsilon = 0$). MVT predictions were determined for each rat; the time spent in each state was taken from a given rat's data. For each model, we fit both group level parameters and individual parameters for each rat using an expectation-maximization algorithm (*Huys et al., 2011*).

To model subjective costs, a free parameter $c$ representing an aversion to leaving the patch was subtracted from the leaving threshold (*Wikenheiser et al., 2013*; *Carter and Redish, 2016*):

$$Q_{cost}(leave) = -c + \gamma(leave) * V_{cost}(s_{first}).$$

To investigate whether nonlinear reward utility could explain rats' overharvesting behavior, we tested models in which the utility of a reward received in the task increased in a sublinear fashion with respect to the magnitude of the reward. Two different utility functions were tested: a power law function and a steeper constant relative risk aversion (CRRA) utility function that became increasingly risk averse with larger rewards, both with free parameter $\eta$:

$$Q_{utility}(stay, s) = U(stay, s) + \gamma(stay, s) * V_{utility}(s_{next})$$

$$U_{power}(stay, s) = R(stay, s)^{\eta}, \text{or}$$

$$U_{CRRA}(stay, s) = \frac{R(stay, s)^{1-\eta} - 1}{1 - \eta}.$$

To examine linear and nonlinear underestimation of post-reward delays, respectively, the time spent in post-reward delay (ITI) states was transformed, with free parameter $\alpha$:

$$T_{post-linear}(s_{ITI}) = \alpha T(s_{ITI}), \text{where } 0 < \alpha < 1, \text{or}$$

$$T_{post-power}(s_{ITI}) = T(s_{ITI})^{\alpha}$$

Similarly, for overestimation of pre-reward delays, the handling time and travel time were transformed:

$$T_{pre-delay}(s_{HT}) = \alpha T(s_{HT}), \text{ and}$$

$$\tau_{pre-delay} = \alpha\tau, \text{ where } \alpha > 1.$$

For the exponential discounting model, β was fit as a free parameter.

As standard hyperbolic discounting cannot conveniently be expressed recursively, this model was implemented using the µAgents model described by **Kurth-Nelson and Redish (2009)**. The value functions of the overall model, $Q^{\mu Agent}(stay, s)$ and $Q^{\mu Agent}(leave)$, were the average of the µAgents, each with their own exponential discount factor $\beta_i$, and thus individual reward functions $R_i(stay, s)$, discount functions $\gamma_i(stay, s)$ and $\gamma_i(leave)$, and value functions $Q_i(stay, s)$, $Q_i(leave)$, and $V_i(s)$:

$$Q_i(stay, s) = R_i(stay, s) + \gamma_i(stay, s) * V_i(s_{next})$$

$$Q^{\mu Agent}(stay, s) = \frac{1}{10}\sum_i R_i(stay, s) + \gamma_i(stay, s) * V_i(s_{next})$$

$$Q_i(leave) = \gamma_i(leave) * V_i(s_{first})$$

$$Q^{\mu Agent}(leave) = \frac{1}{10}\sum_i \gamma_i(leave) * V_i(s_{first})$$

If the µAgent discount factors, $\beta_i$, are drawn from an exponential distribution with rate parameter $\lambda > 0$, the discounting function of the overall model approximated the standard hyperbolic discount function, $reward/(1 + k * delay)$, with discount rate $k = 1/\lambda$. This model was implemented using 10 µAgents with $\beta_i$ equal to the 5%, 15%, ..., 95% quantile of the exponential distribution. The relationship of this implementation of the µAgent model to the standard hyperbolic discount function is presented in **Figure 6**. $k$ was fit as a free parameter.

The constant sensitivity discounting model was based on **Ebert and Prelec (2007)**. In this model, hyperbolic time preferences are produced via exponential discounting with insensitivity to longer delays. To implement this model, insensitivity to all time delays — the decision time, pre-reward delay, reward time, and post-reward delay, and travel time — was achieved using a power function, just as in the nonlinear post-reward delay model. This model was then equivalent to the exponential discounting model, replacing the time in each state $T(s)$ with a power function of the time in each state $T(s)^{\alpha}$.

Quasi-hyperbolic discounting was originally formulated for discrete time applications (**Laibson, 1997**). We used the continuous time formulation from **McClure et al. (2007)**, in which the value functions of the overall model were the weighted sum of two exponential discount systems, a steep discounting β system that prefers immediate rewards and a slower discounting δ system, each with their own reward functions, $R_{\beta}(stay, s)$ and $R_{\delta}(stay, s)$, and discount functions $\gamma_{\beta}(stay, s)$, $\gamma_{\beta}(leave)$, $\gamma_{\delta}(stay, s)$, and $\gamma_{\delta}(leave)$:

$$Q_{\beta}(stay, s) = R_{\beta}(stay, s) + \gamma_{\beta}(stay, s) * V_{\beta}(s_{next})$$

$$Q_{\beta}(leave) = \gamma_{\beta}(leave) * V_{\beta}(s_{first})$$

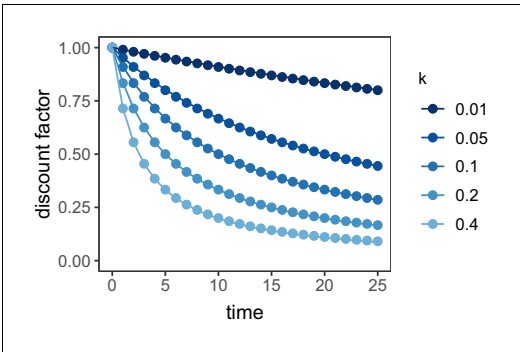

**Figure 6.** Discount function of the µAgent hyperbolic discounting model vs. standard hyperbolic discounting. Points represent the standard hyperbolic discounting function, $1/(1 + k * time)$. Lines represent the µAgent discount function in which the discount factor for each of the 10 µAgents was equal to the 5–95% quantile of an exponential distribution with rate parameter $\lambda = 1/k$.
DOI: https://doi.org/10.7554/eLife.48429.020

$$Q_\delta(stay, s) = R_\delta(stay, s) + \gamma_\delta(stay, s) * V_\delta(s_{next})$$

$$Q_\delta(leave) = \gamma_\delta(leave) * V_\delta(s_{first})$$

The value functions of the overall quasi-hyperbolic discounting model were:

$$Q^{quasi}(stay, s) = \omega * Q_\beta(stay, s) + (1 - \omega) * Q_\delta(stay, s)$$

$$Q^{quasi}(leave) = \omega * Q_\beta(leave) + (1 - \omega) * Q_\delta(leave)$$

where $0 < \omega < 1$ was the weight of the $\beta$ system relative to the $\delta$ system. $\beta$, $\delta$, and $\omega$ were all free parameters.

## Intertemporal choice task models

Similar to the foraging task, events within the intertemporal choice task marked state transitions, and all state transitions were deterministic except for decisions to choose the smaller-sooner option (SS) or larger-later option (LL), which occurred only in decision states (abbreviated state space diagram in *Figure 4—figure supplement 1*). From decision states, animals transitioned to delay, reward, and post-reward delay (ITI) states for the chosen option — the delay, reward and ITI for the SS and LL options were represented by separate states. The value of choosing SS or LL in decision state $s$ is the discounted value of the next state, the following delay state:

$$Q(SS, s) = \gamma(s) * Q(SS\,Delay)$$

$$Q(LL, s) = \gamma(s) * Q(LL\,Delay)$$

The value of delay states were the discounted value of the reward state for that action, the value of reward states were the reward for that action plus the discounted value of the ITI state for that action, and the value of ITI states were the discounted value of the next decision state:

$$Q(SS\,Delay) = \gamma(SS\,Delay) * Q(SS\,Reward)$$

$$Q(SS\,Reward) = R(SS\,Reward) * \gamma(SS\,Reward) * Q(SS\,ITI)$$

$$Q(SS\,ITI) = \gamma(SS\,ITI) * V(s_{next\,dec})$$

where the value of the next decision state, $V(s_{next\,dec})$ is the greater of $Q(SS, s_{next\,dec})$ and $Q(LL, s_{next\,dec})$. Decisions were made assuming the value of $Q(SS, s)$ and $Q(LL, s)$ were represented as Gaussian distributions with noise that scaled with their magnitude. The probability of choosing the LL option was the probability that a random sample from the LL distribution was greater than a random sample from the SS distribution for that state:

$$p(choose\,LL, s) = 1 - \phi\left(\frac{Q(SS, s) - Q(LL, s))}{\sqrt{\sigma^2 * [Q(SS, s)^2 + Q(LL, s)^2)]}}\right)$$

where $\phi$ is the normal cumulative distribution function.

The nonlinear reward utility, biased time perception, and temporal discounting models were implemented as they were in the foraging task.

For the one-trial horizon discounting models, the value of choosing a given option was the discounted value of the reward on the current trial only, with delay $d$ and reward $r$:

$$Q_{exp} = e^{-\beta * d} * r$$

$$Q_{hyp} = \frac{r}{1 + k * d}$$

$$Q_{cs} = e^{-\beta * d^{\alpha}} * r$$

$$Q_{quasi} = [\omega * e^{-\beta * d} + (1 - \omega) * e^{-\delta * d}] * r$$

To calculate the probability of choosing the LL option, the same decision rule was used as in the all-future horizon model.

## Model comparison

All models had two parameters except for the constant sensitivity discounting model with three and the quasi-hyperbolic discounting model with four. To determine the model that provided the best fit to the data, while accounting for the increased flexibility of these models, we calculated the Bayesian Information Criterion over the group level parameters (iBIC) (*MacKay, 2003*; *Huys et al., 2011*). iBIC penalizes the log marginal likelihood, $logp(D \mid \theta)$, which is the integral of the log likelihood of the data $D$ over the distribution of group level parameters $\theta$, for model complexity. Complexity is determined by the number of parameters $k$, and the size of the penalty depends on the total number of observations, $n$:

$$iBIC = logp(D \mid \theta) + \frac{k}{2} log(n).$$

As in *Huys et al. (2011)*, we use a Laplace approximation to the log marginal likelihood:

$$logp(D \mid \theta) = -\frac{n}{2} log(2\pi) * s + \sum_{i=1}^{s} p(D_i \mid \theta_i) p(\theta_i \mid \theta) - \frac{\sum_{i=1}^{s} logdet(Hf(\theta_i))}{2}$$

where $s$ is the number of subjects, and $Hf(\theta_i)$ is the hessian matrix of the likelihood for subject $i$ at the individual parameters $\theta_i$.

To compare the fit of the quasi-hyperbolic discounting model across the foraging and intertemporal choice tasks, a cross-validation method was used. Data from each task was separated into thirds. The quasi-hyperbolic discounting model was fit to 2 of the samples from each task using maximum likelihood estimation (fitting only individual parameters for each rat). The log likelihood of the data from the left out sample was evaluated. This process was repeated three times, leaving out each of the samples once, and we took the sum of the likelihood of the three left out samples. As the structure of variability was different between the foraging model (variability in the patch leaving threshold) and intertemporal choice models (noise in the estimates of SS and LL values), to compare the discount function fit to the foraging task on intertemporal choice data, a new noise parameter was fit to the intertemporal choice data (and vice-versa). We report the difference in the log likelihood of the data using parameters fit to the intertemporal choice task and of the log likelihood using parameters fit to the foraging task (*Figure 5*).

## Acknowledgements

We thank Dr. Gary Aston-Jones for helpful discussions. This work was supported by NIH grant F31MH109286 (GAK) and the Princeton Program in Cognitive Science.

## Additional information

### Funding

| Funder | Grant reference number | Author |
| --- | --- | --- |
| National Institute of Mental Health | F31MH109286 | Gary A Kane<br>Jonathan D Cohen |

The funders had no role in study design, data collection and interpretation, or the decision to submit the work for publication.

### Author contributions

Gary A Kane, Conceptualization, Formal analysis, Investigation, Visualization, Methodology, Writing—original draft, Writing—review and editing; Aaron M Bornstein, Nathaniel D Daw, Conceptualization, Formal analysis, Writing—original draft, Writing—review and editing; Amitai Shenhav, Robert C Wilson, Conceptualization, Writing—original draft, Writing—review and editing; Jonathan D Cohen, Conceptualization, Supervision, Writing—original draft, Writing—review and editing

### Author ORCIDs

Gary A Kane (iD) https://orcid.org/0000-0002-7703-5055
Aaron M Bornstein (iD) http://orcid.org/0000-0001-6251-6000
Robert C Wilson (iD) http://orcid.org/0000-0002-2963-2971
Nathaniel D Daw (iD) http://orcid.org/0000-0001-5029-1430

### Ethics

Animal experimentation: This study was performed in strict accordance with the recommendations in the Guide for the Care and Use of Laboratory Animals of the National Institutes of Health. All procedures were approved by the Princeton University (Protocol 1969) and Rutgers University (Protocol 14-075) Institutional Animal Care and Use Committees.

### Decision letter and Author response

Decision letter https://doi.org/10.7554/eLife.48429.025
Author response https://doi.org/10.7554/eLife.48429.026

## Additional files

### Supplementary files

• Source code 1. Foraging mixed effects models.
DOI: https://doi.org/10.7554/eLife.48429.021

• Source code 2. Intertemporal choice mixed effects model.
DOI: https://doi.org/10.7554/eLife.48429.022

• Transparent reporting form
DOI: https://doi.org/10.7554/eLife.48429.023

### Data availability

All data generated or analysed during this study are included in the manuscript and supporting files.

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
