## [Decision Letter]

Thank you for submitting your article "Rats exhibit similar biases in foraging and intertemporal choice tasks" for consideration by *eLife*. Your article has been reviewed by three peer reviewers, and the evaluation has been overseen by Geoffrey Schoenbaum as the Reviewing Editor and Laura Colgin as the Senior Editor. The following individuals involved in review of your submission have agreed to reveal their identity: Vijay Namboordiri (Reviewer #2); Marijn van Wingerden (Reviewer #3).

The reviewers have discussed the reviews with one another and the Reviewing Editor has drafted this decision to help you prepare a revised submission.

Summary:

This study aims to reconcile behavior exhibited in patch foraging tasks with that shown during intertemporal choice. Across several experiments, the authors replicate the "overharvesting" biases observed previously, examining several competing proposals to explain them, concluding that models incorporating time preferences are required for any adequate explanation. The quasi hyperbolic models derived from patch foraging work were best to describe behavior in both tasks, suggesting these models may represent a general underlying function in both behaviors.

Essential revisions:

The reviewers were in agreement that the work was well done and addressed an important question. Each had concerns and questions; however, it was agreed that as long as the authors make a good faith attempt to address these, none of them required a very particular answer for the paper to be acceptable for publication. It is important however that they are carefully considered and adequately addressed. These include, in particular, discussion of how the current proposal relates to prior ideas as well as other issues raised (please refer to individual reviews below). Despite these concerns, the reviewers were overall impressed with the work and its contribution to the literature.

Reviewer #1:

Summary:

The authors test rats' temporal decision-making in six tasks: five foraging tasks and one intertemporal choice task. The foraging tasks test increasingly complex predictions of marginal value theory (MVT). In the first, second, and fifth tasks, they systematically validate the qualitative predictions that rats will stay longer in patches that i) deliver larger reward, ii) are separated by longer travel times, iii) deplete at slower rates, iv) deliver reward at shorter intervals. However, they observed that rats 'overharvested': stayed longer in patches than is optimal and predicted by MVT. These findings add to a growing literature refuting MVT by reporting overharvesting. In the 'scale' task, the authors observe that when all magnitudes and rewards are scaled 2x, the rats overharvested longer than the 1x condition. In the 'pre-vs-post' task, the authors replicate divergent effects of time requirements placed before or after rewards (a common feature of experiments showing suboptimal intertemporal choice) layered on top of foraging decisions. The fifth task refutes any hypothesis that attributes suboptimal behavior to failure to consider post-reward delays. The authors further test rats in more traditional intertemporal choice experiments so that they can compare behavior across all tasks and find the best-fitting quantitative model. Interestingly, they find a two-parameter model they call 'quasi-hyperbolic' to fit better than any one-parameter model attempted.

Strengths:

This is a relatively thorough investigation of the sensitivities of rats behavior to many of the variables that could possibly impact optimal foraging behavior. For every variable investigated, the authors observe the expected qualitative effect predicted by MVT, and replicate qualitative suboptimalities observed in simpler choice contexts within the foraging context. The authors also have the opportunity to investigate the relationship across behaviors for each individual.

Weaknesses in analysis:

In the Scale Experiment, did the rats overharvest 2x as long in the 2x condition? Perhaps this shouldn't be presented as unexpected.

The best-fit evaluation (quasi-hyperbolic) is the only 2-parameter version. The authors should invent additional 2-parameter models for further comparison.

The variability across rats in identified parameters should be better discussed and illustrated. The analysis is not a convincing illustration that rats' performance in one task is predictive of performance in another. Perhaps a correlation in parameters across rats would be more convincing.

Only a few specific forms of misrepresenting time spent before and after reward are tested, but this is presented as a thorough refutation of theories of altered time perception.

Weaknesses in presentation:

The numerous motivating hypotheses describing decision making are poorly presented. I recommend a table describing them and indicating what they predict for each of the experiments. A glossary clearly explaining each theory would also be helpful. Why isn't the fifth foraging experiment included in Table 1? Nonlinear reward magnitude utility is described as "decreasing marginal value," which is confusing.

Reviewer #2:

This manuscript by Kane et al. tests behavior of rats in a foraging-like task and a delay-discounting task to advance the claim that behavior in both of these tasks can be explained by a single set of rules. This is an important claim since prior papers have argued that both humans and animals might show different behaviors depending on the nature of the tasks, even though value reduction of a delayed reward should play a role across both conditions. My only major concern relates to the treatment of prior work.

1) The authors should cite Seinstra, Sellitto and Kalenscher, 2017. This paper argues that human behavior across foraging and delay-discounting settings can be explained by a single framework. The framework is rate maximization and hence, is different from the one advanced by the authors. But nevertheless, it would be good to discuss this paper since it is highly relevant.

2) The argument presented in Blanchard et al., 2013 is that post reward delays are not accounted for, primarily due to the deficit in learning these delays. This is because most post reward delays are not signaled explicitly in the environment. Therefore, they show that adding additional small rewards after the post-reward delay is sufficient to cause learning of the delay and for inclusion of the delay in the decision process. This is different from the interpretation presented here. Related to this, I am not convinced that the rats learn the post-reward delays during the foraging task. The ITI is adjusted depending on the DT, but if the animals are not keeping track of time since trial start, they would not learn this structure of the environment. Of course, in the absence of such learning, it is not surprising that post-reward delays are not weighted equally as the pre-reward delays. So, one possible modification could be that only a weighted value of the post-reward delay is included in the decision.

3) Another set of papers actually also provide a theoretical framework to explain intertemporal decision-making during foraging and delay-discounting. I will confess that I am an author of these papers. While I typically am loathe to suggesting my own papers during peer reviews, I think it is highly relevant in this case. The papers introducing this model are Namboodiri et al., 2014 A general theory of intertemporal decision-making and the perception of time, Frontiers in Beh. Neurosci., (primary research article) and Namboodiri et al., 2014 Rationalizing decision-making: understanding the cost and perception of time, Timing and Time Perception Reviews, (review). There are a few other papers investigating errors in decision-making using a trial-by-trial approach, but the mean field model presented in these papers is sufficient to highlight the relevance. In these papers, the main model that is presented for illustration purposes is almost a toy model, and was falsified by Carter and Redish, 2016. However, there are simple and more realistic extensions that were not considered by Carter and Redish and are in the Appendix of the original paper. I will present a (hopefully short) summary of its application to the foraging data presented here to show that at least all of the qualitative findings hold within a rate maximization perspective over a limited temporal horizon.

Decision-making in this TIMERR model is based on comparing the expected reward rate over a temporal horizon (T_ime_) and the expected delays to the next reward. Of course, even though the post-reward delay is after the delay to the next reward, it should be incorporated in the decision, if learned. If one assumes a potential learning deficit in these tasks due to the delay not being explicitly signaled, a simple model for their inclusion could be a weight (w) multiplied by the perceived delay (assumed to be the objective delay, t_post_, for simplicity). So effectively, the time horizon of consideration is T_ime_+t_pre_+w*t_post_. If the animal estimates an average reward rate (a_est_) to be obtained during the temporal horizon Time, the net reward expected over Time is a_est_*T_ime_. The toy model in the above papers assumes that a_est_ is calculated over a time window exactly equal to T_ime_, but this need not be the case. This is the key modification that Carter and Redish, 2016, did not consider. Thus, the expected reward rate for the nth visit to a patch equals

RR(n) = (r-d*n+a_est_*T_ime_)/(t_decision_+t_pre_+w*t_post_+T_ime_)

Where r is the starting reward, d is the rate of reduction of the reward and t_decision_ is the time until the decision is made after the cue. Since t_harvest_=t_decision_+t_pre_+t_post_, the above can be written as

RR(n) = (r-d*n+a_est_*T_ime_)/(t_harvest_-(1-w)*t_post_+T_ime_)

Similarly, the expected reward rate for the limited temporal horizon until the next reward after traveling to a new patch is

RR(travel) = (r+a_est_*T_ime_)/(t_travel_+t_harvest_-(1-w)*t_post_+T_ime_)

Thus, the number of visits one would stay in a patch would be the number at which the above two expected reward rates equal. Doing some algebra and noting that a_est_ is actually dependent on the task itself (average estimated reward rate over some time horizon prior to the decision), hopefully, it will be clear that the findings of this manuscript at least qualitatively are consistent with this model. There are three free parameters here, viz., T_ime_, w, and the time horizon over which a_est_ is calculated. In the interest of brevity of this review, I will not prolong this, but the main point is that this simple model of constrained rate maximization is at least qualitatively consistent with the key findings of this manuscript. Indeed, a more accurate model would require considering potential biases due to errors in time perception as treated in Namboodiri et al. 2014, Analytical calculation of errors in time and value perception due to a subjective time accumulator: A mechanistic model and the generation of Weber's law, Neural Computation, 2016.

I want to highlight that the point of my raising this is not to force a detailed consideration of my own work. I raise it only because the authors genuinely seem to care about these problems in decision-making and may perhaps find a different perspective on their data valuable.

4) Related to points 2 and 3 above, the assumption is that animals have learned the structure of the task fully and can include all appropriate values objectively in their decision-making. This is an assumption that may be worth explicitly pointing out.

5) Another descriptive discounting function that could potentially also fit the data is Killeen, 2009. This form is more of a descriptive model than a normative one, but has some math that is superficially similar to the TIMERR model, and would also likely be consistent with these data.

Reviewer #3:

The manuscript by Kane et al. is an exciting addition to the literature than has been trying to reconcile animal decision making in patch foraging vs. intertemporal choice problems. Using a surprisingly low number of animals, the authors robustly replicate a number of decision biases that have been shown in previous research, collectively described as "overharvesting", related to handling time, pre/post-delay sensitivity and other choice attributes. Next, they proceed to examine several (competing) proposals for cognitive choice preferences that could explain some of these biases. The authors convincingly demonstrate that any model not assuming time preferences fails to adequate explain all choice biases displayed by the animals, and that of the models incorporating time preferences, β-δ quasi-hyperbolic discounting provides the best fit across the paradigms, and also in almost all of the single experiment cases. Importantly, the quasi hyperbolic models derived from patch foraging work well to describe behavior in the intertemporal choice setting and vice versa, suggesting that they might touch upon a basic choice heuristic that plays a significant role animal decision making.

The manuscript is well written and concise. I have no objections to the line of argument in general. One concern that I hope the authors can address concerns the following: in the Markov analyses, future rewards are taken into account and are assumed to include all rewards up to the last one in the current block, to induce the incorporation of the long-term reward rate into the decisions. It is however unclear why the authors chose to expand the horizon to the last choice. It seems to me that the "reward horizon" of animals might actually be limited or even fluctuate, for example based on satiety. It seems that with the current data at hand, it would be possible to model the reward horizon as a free parameter to approach this choice empirically. What horizon appears from this analysis? And can the authors replicate the "failure" for cross-explanation between foraging and ITC when the horizon is set to only the current trial?

Additionally, while the quasi-hyperbolic discount model provides an overall better fit than for example the diminishing marginal returns model, it seems that animals exhibit both time preferences AND experience sensitivity to diminishing marginal returns, based on reward size differences and (the very real consequences of) already accumulated rewards.

If accumulated marginal returns are not important, one would assume that a median split (early vs. late) on the trials in the foraging dataset, or the first 5 vs. last 5 trials in the ITC dataset would produce similar model estimates. However, if the model fits diverge substantially this would suggest that an extension of the model with quasi-hyperbolic time-preferences to include marginal reward size/accumulation effects might provide an even better fit. I think this question could also be answered with the current dataset that is unique equipped to compare models under different experimental circumstances.

---

## [Author Response]

Essential revisions:The reviewers were in agreement that the work was well done and addressed an important question. Each had concerns and questions; however, it was agreed that as long as the authors make a good faith attempt to address these, none of them required a very particular answer for the paper to be acceptable for publication. It is important however that they are carefully considered and adequately addressed. These include, in particular, discussion of how the current proposal relates to prior ideas as well as other issues raised (please refer to individual reviews below). Despite these concerns, the reviewers were overall impressed with the work and its contribution to the literature.Reviewer #1:Strengths:This is a relatively thorough investigation of the sensitivities of rats behavior to many of the variables that could possibly impact optimal foraging behavior. For every variable investigated, the authors observe the expected qualitative effect predicted by MVT, and replicate qualitative suboptimalities observed in simpler choice contexts within the foraging context. The authors also have the opportunity to investigate the relationship across behaviors for each individual.Weaknesses in analysis:In the Scale Experiment, did the rats overharvest 2x as long in the 2x condition? Perhaps this shouldn't be presented as unexpected.

We apologize for not being clear with regard to the interpretation of statistical analysis. Rats did not overharvest by a factor of two, rather they overharvested for an additional 2 trials. This point has been clarified in the text by adding units to regression coefficients where appropriate (e.g. “β = 1.937 trials, SE = .193…”). Further, we now clarify that this is not being presented as surprising or unexpected, but only as contrary to predictions of the Marginal Value Theorem:

“[Rats] overharvested in both A and B conditions… and, contrary to predictions of MVT, they stayed in patches significantly longer….”

The best-fit evaluation (quasi-hyperbolic) is the only 2-parameter version. The authors should invent additional 2-parameter models for further comparison.

As suggested by reviewers 2 and 3 (comments 6 and 3 respectively), we tested additional models with greater complexity (> 2 parameters), including a constant time sensitivity model that combined exponential discounting with biases in time perception, and a model reminiscent of additive-utility discounting, combining nonlinear reward utility, biased time perception and exponential discounting (see below). Overall, neither of these models performed better than the quasi-hyperbolic discounting model. We would gladly test any additional models the reviewer may suggest.

In addition, to account for different numbers of parameters across models, we used the Bayesian Information Criterion which penalizes the fit of a model for its complexity (i.e. the number of parameters), and thus, does not provide a particular advantage to a model with more parameters. In fact, because BIC uses a fixed penalty per additional parameter, it may “overpenalize” more complex models in situations where an additional parameter does not provide a multiplicative increase in explanatory power (Bernardo and Smith, 1994, Bayesian Theory).

The variability across rats in identified parameters should be better discussed and illustrated. The analysis is not a convincing illustration that rats' performance in one task is predictive of performance in another. Perhaps a correlation in parameters across rats would be more convincing.

We agree that investigating correlations between parameters could provide an additional line of evidence that rats behave in similar ways across the two tasks, but we were wary of this analysis for two reasons: i) correlations between model parameters do not imply that behavior in one task can predict behavior in the other task and ii) with only 8 rats in each experiment, correlations are unreliable (e.g. see Schonbrodt and Perugini, 2013, J Res Personality) and greatly underpowered. A power analysis revealed that a correlation coefficient r = 0.833 would be needed to achieve power = 0.8 with significance level of p=0.05.

Despite these issues, we conducted a few exploratory correlation analyses. First, we examined the correlation between the behavioral effect of the post-reward delay across tasks (i.e. the effect of the delay in the Post-Reward Delay foraging experiment, and the interaction between the effect of the LL delay and task condition – the difference in the effect of the LL delay when it affected reward rate vs. when it did not affect reward rate). These effects were estimated for each rat using mixed effects models. There was a strong, but not significant, correlation between the effect of the delay (r = .606, p = .111).

We also tested correlations between each of the four quasi-hyperbolic model parameters across the Post-Reward Delay foraging task and intertemporal choice task. Again, correlations between some parameters were quite strong, but not significant following Bonferroni correction for multiple comparisons (see Author response image 1). Although these observations must be treated with caution, they suggest there may be stability of behavior and model parameters across tasks.

Only a few specific forms of misrepresenting time spent before and after reward are tested, but this is presented as a thorough refutation of theories of altered time perception.

We regret that we gave this impression, as we certainly do not mean to suggest that this work is a thorough refutation that altered time perception contributes to overharvesting or suboptimal behavior in intertemporal choice tasks. Rather, our goal is to communicate that a particular set of models that represent the hypothesis that animals are differently sensitive to pre- vs. post-reward delays do not explain rat foraging or intertemporal choice data as well as the quasi-hyperbolic discounting model. We have revised the manuscript to provide more context for these hypotheses by providing more discussion of Blanchard et al., 2013 (as suggested by reviewer 2). We emphasize the behavioral sensitivity to post-reward delays in the Post-Reward Delay experiment:

“Prior studies of intertemporal choice behavior suggest that animals are insensitive to post-reward delays, suggesting that they are only concerned with maximizing short-term reward rate (Stephens and Anderson, 2001, Bateson and Kacelnik 1996) or they may not have learned the duration of post-reward delays, and underestimate this duration in their decision process (Pearson et al., 2010, Blanchard and Hayden, 2013). […] However, it is possible that this finding could be explained by other forms of altered time perception that remain to be described.”

And in the discussion, we clarify that our findings do not implicate quasi-hyperbolic discounting as the one-and-only explanation for suboptimal behavior in foraging and intertemporal choice tasks:

“Although quasi-hyperbolic discounting provided the best singular explanation for rat behavior across our tasks, many of the other models tested were capable of explaining some of the biases exhibited by rats. […] Importantly, our data indicate that quasi-hyperbolic discounting may provide a link between foraging and intertemporal choice tasks, and it highlights the importance of future work considering the source of time preferences.”

Weaknesses in presentation:The numerous motivating hypotheses describing decision making are poorly presented. I recommend a table describing them and indicating what they predict for each of the experiments. A glossary clearly explaining each theory would also be helpful. Why isn't the fifth foraging experiment included in Table 1? Nonlinear reward magnitude utility is described as "decreasing marginal value," which is confusing.

Thank you for the valuable suggestion to include a table of each hypothesis along with predictions for each experiment. The table is now included and cited in the first paragraph of the Results section: “Quasi-hyperbolic discounting best explains behavior across all tasks.”

We expanded this table to include the fifth experiment. The fifth foraging experiment (post-reward delay experiment) was not initially included in the table because a separate cohort of rats was tested and we did not adjust the post-reward delay to control trial time as was done for the first four experiments. These differences are now noted in the table and its caption.

To avoid confusion, we changed “diminishing marginal value” and “diminishing marginal returns” to “nonlinear reward utility” throughout the text.

Reviewer #2:This manuscript by Kane et al. tests behavior of rats in a foraging-like task and a delay-discounting task to advance the claim that behavior in both of these tasks can be explained by a single set of rules. This is an important claim since prior papers have argued that both humans and animals might show different behaviors depending on the nature of the tasks, even though value reduction of a delayed reward should play a role across both conditions. My only major concern relates to the treatment of prior work.1) The authors should cite Seinstra, Sellitto and Kalenscher, 2017. This paper argues that human behavior across foraging and delay-discounting settings can be explained by a single framework. The framework is rate maximization and hence, is different from the one advanced by the authors. But nevertheless, it would be good to discuss this paper since it is highly relevant.

We agree that this paper is highly relevant to the current work and have included a brief discussion of these findings:

“[The idea that decision-making systems evolved to solve foraging problems rather two-alternative choice problems] has been challenged by a recent study of human decision-making in foraging and intertemporal choice tasks, finding that a long-term rate maximization model explained behavior in both tasks better than a standard hyperbolic discounting model (Seinstra et al., 2017). Results from the present study support the interpretation that foraging and intertemporal choice behavior can be explained by a common model, but suggest that this model is a form of discounted rate maximization.”

2) The argument presented in Blanchard et al., 2013 is that post reward delays are not accounted for, primarily due to the deficit in learning these delays. This is because most post reward delays are not signaled explicitly in the environment. Therefore, they show that adding additional small rewards after the post-reward delay is sufficient to cause learning of the delay and for inclusion of the delay in the decision process. This is different from the interpretation presented here.

We agree with the reviewer’s interpretation of Blanchard et al., 2013 and apologize for inadvertently misrepresenting their argument. We have revised the way in which we discuss these findings throughout the paper (below), but this revised discussion does not change our interpretation of the results.

In the Introduction: “An alternative explanation for the preference for immediate rewards in intertemporal choice tasks is that animals simply underestimate the duration of post-reward delays; that is, delays between receiving reward and making the next decision (Blanchard et al., 2013, Pearson et al., 2010). […] Consistent with this hypothesis, providing an explicit cue for the duration of the post-reward delay or increasing its salience by providing a small reward at the end of the post-reward delay reduces temporal discounting (Blanchard et al., 2013, Pearson et al., 2010).”

Also, in the Results (see reviewer 1, comment 4), and in the Discussion:

“A number of studies have found that the preference for smaller, more immediate rewards can be explained by insensitivity to post-reward delays (Bateson et al., 1996, Blanchard et al., 2013, Pearson et al., 2010, Stephens and Anderson 2001, Mazur, 1991). One hypothesis for why animals fail to incorporate post-reward delays into decisions is that they haven't learned the structure of the task well, and thus cannot accurately predict future post-reward delays. It has been shown that providing explicit cues for the post-reward delays or increasing the salience of post-reward delays helps animals incorporate these delays into their decisions, reducing the bias towards selecting smaller, more immediate rewards over larger, delayed ones (Blanchard et al., 2013, Pearson et al., 2010).”

Related to this, I am not convinced that the rats learn the post-reward delays during the foraging task. The ITI is adjusted depending on the DT, but if the animals are not keeping track of time since trial start, they would not learn this structure of the environment. Of course, in the absence of such learning, it is not surprising that post-reward delays are not weighted equally as the pre-reward delays. So, one possible modification could be that only a weighted value of the post-reward delay is included in the decision.

Results from the Post-Reward Delay Experiment, which directly tested rats’ ability to incorporate the post-reward delay into decisions, showed that rats were sensitive to post-reward delays. However, in this experiment, the post-reward delay was fixed, whereas in the other foraging experiments, the post-reward delay was variable. It’s possible that rats struggled to incorporate the variable post-reward delay into decision processes in other experiments. To test this hypothesis, we tested the suggested model, which included a weighted value of the post-reward delay (referred to as the linear underestimation of post-reward delays model; formal definition).

Figure 3—figure supplement 3 shows that this model, when fit to rat behavior, predicts that rats will leave patches earlier with a longer pre-reward delay and shorter post-reward delay. However, according to Bayes Information Criterion (Figure 3—figure supplement 5), it does not fit as well as the quasi-hyperbolic discounting model in this experiment.

3) Another set of papers actually also provide a theoretical framework to explain intertemporal decision-making during foraging and delay-discounting. I will confess that I am an author of these papers. While I typically am loathe to suggesting my own papers during peer reviews, I think it is highly relevant in this case. The papers introducing this model are Namboodiri et al., 2014 A general theory of intertemporal decision-making and the perception of time, Frontiers in Beh. Neurosci., (primary research article) and Namboodiri et al., 2014 Rationalizing decision-making: understanding the cost and perception of time, Timing and Time Perception Reviews, (review). There are a few other papers investigating errors in decision-making using a trial-by-trial approach, but the mean field model presented in these papers is sufficient to highlight the relevance. In these papers, the main model that is presented for illustration purposes is almost a toy model, and was falsified by Carter and Redish, 2016. However, there are simple and more realistic extensions that were not considered by Carter and Redish and are in the Appendix of the original paper. I will present a (hopefully short) summary of its application to the foraging data presented here to show that at least all of the qualitative findings hold within a rate maximization perspective over a limited temporal horizon.Decision-making in this TIMERR model is based on comparing the expected reward rate over a temporal horizon (T_ime_) and the expected delays to the next reward. Of course, even though the post-reward delay is after the delay to the next reward, it should be incorporated in the decision, if learned. If one assumes a potential learning deficit in these tasks due to the delay not being explicitly signaled, a simple model for their inclusion could be a weight (w) multiplied by the perceived delay (assumed to be the objective delay, t_post_, for simplicity). So effectively, the time horizon of consideration is T_ime_+t_pre_+w*t_post_. If the animal estimates an average reward rate (a_est_) to be obtained during the temporal horizon Time, the net reward expected over Time is a_est_*T_ime_. The toy model in the above papers assumes that a_est_ is calculated over a time window exactly equal to T_ime_, but this need not be the case. This is the key modification that Carter and Redish, 2016, did not consider. Thus, the expected reward rate for the nth visit to a patch equalsRR(n) = (r-d*n+a_est_*T_ime_)/(t_decision_+t_pre_+w*t_post_+T_ime_)Where r is the starting reward, d is the rate of reduction of the reward and t_decision_ is the time until the decision is made after the cue. Since t_harvest_=t_decision_+t_pre_+t_post_, the above can be written asRR(n) = (r-d*n+a_est_*T_ime_)/(t_harvest_-(1-w)*t_post_+T_ime_)Similarly, the expected reward rate for the limited temporal horizon until the next reward after traveling to a new patch isRR(travel) = (r+a_est_*T_ime_)/(t_travel_+t_harvest_-(1-w)*t_post_+T_ime_)Thus, the number of visits one would stay in a patch would be the number at which the above two expected reward rates equal. Doing some algebra and noting that a_est_ is actually dependent on the task itself (average estimated reward rate over some time horizon prior to the decision), hopefully, it will be clear that the findings of this manuscript at least qualitatively are consistent with this model. There are three free parameters here, viz., T_ime_, w, and the time horizon over which a_est_ is calculated. In the interest of brevity of this review, I will not prolong this, but the main point is that this simple model of constrained rate maximization is at least qualitatively consistent with the key findings of this manuscript. Indeed, a more accurate model would require considering potential biases due to errors in time perception as treated in Namboodiri et al., Analytical calculation of errors in time and value perception due to a subjective time accumulator: A mechanistic model and the generation of Weber's law, Neural Computation, 2016.I want to highlight that the point of my raising this is not to force a detailed consideration of my own work. I raise it only because the authors genuinely seem to care about these problems in decision-making and may perhaps find a different perspective on their data valuable.

We greatly appreciate the reviewer's thoughtful consideration of these models. We agree that the TIMERR model is very relevant to this manuscript and we have now included a brief discussion of the TIMERR model:

“Similarly, short-term maximization rules predict that animals seek to maximize reward over shorter time horizons; this may also be motivated as an approximation to long-term reward maximization as it may be difficult to accurately predict all future rewards (Stephens 2002, 2004). Along similar lines, Namboodiri et al., 2014, argue that, rather than maximizing long-term reward rate into the future, animals may select options that maximize reward rate up to the current point in time, or due to environmental factors (e.g. non-stationarity) or biological constraints (e.g. computational constraints), over a finite interval of time. Just as hyperbolic discounting may arise from imperfect foresight (Gabaix and Laibson, 2017), maximizing reward rate over shorter time horizons predicts hyperbolic time preferences (Namboodiri et al., 2014).”

Additionally, the suggested extension to the TIMERR model is very interesting, but we did not test this model directly for a few reasons. First, this model is difficult to translate into the semi-markov decision process used for all other models. Importantly, the semi-markov process enabled us to accurately model all timing features of the task, including slow delivery of larger rewards that occurred over up to 2.5 s and were consumed over the course of ~5 s. Second, as the TIMERR model predicts hyperbolic time preferences (Namboodiri et al., 2014), we view the additional flexibility of the extended TIMERR model similar to the additional flexibility offered by the constant sensitivity discounting function (a new model included in the manuscript, see comment 6 below) or the quasi-hyperbolic discounting function.

4) Related to points 2 and 3 above, the assumption is that animals have learned the structure of the task fully and can include all appropriate values objectively in their decision-making. This is an assumption that may be worth explicitly pointing out.

We agree that this assumption should be made explicit, and have now done so:

“These models consisted of a set of states that represented the time between each event in each of the tasks (e.g. cues turning on/off, lever press, reward delivery; for state space diagrams of both tasks, see Figure 4—figure supplement 1 and Figure 4—figure supplement 1). These models assumed that animals have learned the appropriate structure of the task (i.e. the time spent and reward obtained in each state) unless otherwise noted.”

5) Another descriptive discounting function that could potentially also fit the data is Killeen, An additive-utility model of delay discounting, Psychol Rev, 2009. This form is more of a descriptive model than a normative one, but has some math that is superficially similar to the TIMERR model, and would also likely be consistent with these data.

We agree that this is an interesting model given our data and thank the reviewer for bringing this model to our attention. This additive-utility discounting model cannot be expressed recursively within the framework of our semi-markov models, but we have tested two new models consisting of the main components of the additive-utility discounting model. The first is the constant sensitivity discounting model from Eberle and Prelec, 2007, which is discussed in Killeen, 2009. This model predicts hyperbolic time preferences via exponential discounting with a bias in time sensitivity (agents are less sensitive to longer delays). The second model combined the main components of the additive utility discounting function – discounted utility instead of value, and decreasing sensitivity to delays – by adding a nonlinear reward utility parameter to the constant sensitivity discounting model.

The constant sensitivity discounting performed well on the foraging task, but did not fit intertemporal choice task data well. The utility + constant sensitivity model performed similarly, but had a higher iBIC than the constant sensitivity discounting model alone (foraging: sum iBIC_without-util_ = 25479, sum iBIC_with-util_ = 25621; intertemporal choice: iBIC_without-util_ = 18021, iBIC_with-util_ = 18024). Because adding the utility parameter did not improve the fit of this model, we have opted to not include the utility version in the manuscript. The constant sensitivity model has been included (the “disc-cs” model). Please see the following figures (sum of iBIC in foraging tasks in Figure 3F, iBIC in intertemporal choice task from Figure 4B).

In addition, the non-linear post-reward delay model has been changed from a exponential decay function to a power function (the same power function used to model biased time perception in the constant sensitivity discounting model). This change was made for two reasons: for consistency with the constant-sensitivity discounting model and the power delay function had a lower iBIC (iBIC_exp_ = 26595.25; iBIC_pwr_ = 26229.35).

Reviewer #3:The manuscript by Kane et al. is an exciting addition to the literature than has been trying to reconcile animal decision making in patch foraging vs. intertemporal choice problems. Using a surprisingly low number of animals, the authors robustly replicate a number of decision biases that have been shown in previous research, collectively described as "overharvesting", related to handling time, pre/postdelay sensitivity and other choice attributes. Next, they proceed to examine several (competing) proposals for cognitive choice preferences that could explain some of these biases. The authors convincingly demonstrate that any model not assuming time preferences fails to adequate explain all choice biases displayed by the animals, and that of the models incorporating time preferences, β-δ quasi-hyperbolic discounting provides the best fit across the paradigms, and also in almost all of the single experiment cases. Importantly, the quasi hyperbolic models derived from patch foraging work well to describe behavior in the intertemporal choice setting and vice versa, suggesting that they might touch upon a basic choice heuristic that plays a significant role animal decision making.The manuscript is well written and concise. I have no objections to the line of argument in general. One concern that I hope the authors can address concerns the following: in the Markov analyses, future rewards are taken into account and are assumed to include all rewards up to the last one in the current block, to induce the incorporation of the long-term reward rate into the decisions. It is however unclear why the authors chose to expand the horizon to the last choice. It seems to me that the "reward horizon" of animals might actually be limited or even fluctuate, for example based on satiety. It seems that with the current data at hand, it would be possible to model the reward horizon as a free parameter to approach this choice empirically. What horizon appears from this analysis?

To improve consistency between the horizon of the foraging and intertemporal choice models, the intertemporal choice model has been modified to include all future rewards, including those on future episodes (in the initial version of the manuscript, this model only considered rewards on the current episode). However, for temporal discounting models, the effective horizon is determined by the rate of discounting – at some point, the weight on future rewards approaches zero, and animals will not consider rewards beyond that time point. Whether reward horizon fluctuates due to satiety or other factors is an interesting question but we unfortunately could not test that directly in the current study.

And can the authors replicate the "failure" for cross-explanation between foraging and ITC when the horizon is set to only the current trial?

We have included a new section in the results comparing one-trial horizon discounting models with the future-rewards discounting models. We performed two comparisons: for all discounting functions, we found that future-rewards models provided a better fit (lower iBIC) and smaller estimates of discount factors relative to one-trial horizon models (see Figure 4—figure supplement 2). However, as asked by the reviewer, we could not replicate the “failure” for cross-explanation between the foraging and ITC tasks due to a shorter horizon. For the quasi-hyperbolic discounting function, we found that the future-rewards model parameters fit to intertemporal choice task did not predict foraging behavior better than one-trial horizon parameters fit to the intertemporal choice task:

“With temporal discounting models that consider all future rewards, the more flexible quasi-hyperbolic discounting function provided the best fit to behavior across tasks. […] Overall, using full horizon temporal discounting models explained more of the intertemporal choice data, produced smaller estimates of discounting factors, but in the present study, it did not improve the ability of a model fit to intertemporal choice data to predict foraging behavior.”

Also, in the Discussion:

“… we found that adding the value of future rewards to intertemporal choice models reduces estimates of discount factors. […] In these cases, obtaining smaller, potentially more accurate estimates of discount factors by including all future rewards into intertemporal choice models may improve cross-task predictions.”

Additionally, while the quasi-hyperbolic discount model provides an overall better fit than for example the diminishing marginal returns model, it seems that animals exhibit both time preferences AND experience sensitivity to diminishing marginal returns, based on reward size differences and (the very real consequences of) already accumulated rewards. If accumulated marginal returns are not important, one would assume that a median split (early vs. late) on the trials in the foraging dataset, or the first 5 vs. last 5 trials in the ITC dataset would produce similar model estimates. However, if the model fits diverge substantially this would suggest that an extension of the model with quasi-hyperbolic time-preferences to include marginal reward size/accumulation effects might provide an even better fit. I think this question could also be answered with the current dataset that is unique equipped to compare models under different experimental circumstances.

In both models, values are calculated on future projections alone; prior accumulated rewards will not have an effect on model predictions. To account for the magnitude of future expected rewards, we have modified both the foraging and intertemporal choice models such that noise in decisions scales with the value estimates. In their current form, the greater the projected future rewards, the more noise in decisions. Consequently, these models can predict that rats will be more likely to select the LL option in the intertemporal choice task as they approach the end of an episode (assuming rewards from future games are highly discounted).

To directly test whether behavior changed between early/late periods in the foraging task or between the first 5/last 5 choices of each episode in the intertemporal choice task, we added a parameter to each of the mixed effects models. Neither the early/late parameter in the foraging mixed effects model nor the first 5/ last 5 parameter in the intertemporal choice mixed effects model had a significant impact (foraging early/late: β = .013, SE = .192, t(7.054) = .069, p = .947; intertemporal choice: β = .018, SE = .042, z = .439, p = .661). We also tested a nonlinear reward utility + quasi-hyperbolic discounting model. Adding the utility parameter did not improve the fit of the quasi-hyperbolic model to either foraging or intertemporal choice data (foraging: sum of iBIC_quasi_ = 25383.31, sum of iBIC_util+quasi_ = 25408.36; intertemporal choice: iBIC_quasi_ = 17981.98, iBIC_util+quasi_ = 17983.86). These analyses suggest that there were no effects of reward accumulation over the course of a foraging session or intertemporal choice episode.